# Affinity-guided labeling reveals P2X7 nanoscale membrane redistribution during BV2 microglial activation

**Benoit Arnould[1‡], Adeline Martz[1†], Pauline Belzanne[2†], Francisco Andrés Peralta[1,3], Federico Cevoli[1], Volodya Hovhannisyan[4], Yannick Goumon[4], Eric Hosy[2], Alexandre Specht[5], Thomas Grutter[1,3]\***

[1]Laboratoire de Chémo-Biologie Synthétique et Thérapeutique (CBST) UMR 7199, équipe Ingénierie Canaux Ioniques, Centre National de la Recherche Scientifique, Université de Strasbourg, Faculté de Pharmacie, Illkirch, France; [2]Interdisciplinary Institute for Neuroscience, CNRS, Université de Bordeaux, Bordeaux, France; [3]University of Strasbourg Institute for Advanced Studies (USIAS), Strasbourg, France; [4]Centre National de la Recherche Scientifique-Unité Propre de Recherche (CNRS-UPR) 3212, Institut des Neurosciences Cellulaires et Intégratives, Strasbourg, France; [5]Laboratoire de Chémo-Biologie Synthétique et Thérapeutique (CBST) UMR 7199, équipe NanoParticules Intelligentes, Centre National de la Recherche Scientifique, Université de Strasbourg, Strasbourg, France

**\*For correspondence:**
grutter@unistra.fr

†These authors contributed equally to this work

**Present address:** ‡CNRS UMR9187, INSERM U1196, Institut Curie, PSL Research University, Orsay, France

## eLife Assessment

The authors employ an unbiased, affinity-guided reagent to label P2X7 receptor and use super-resolution imaging to monitor P2X7 redistribution in response to inflammatory signaling. The evidence is **convincing** and the study will be **valuable** to those studying the dynamics of receptor distribution and clustering.

**Abstract** ATP-gated purinergic P2X7 receptors are crucial ion channels involved in inflammation. They sense abnormal ATP release during stress or injury and are considered promising clinical targets for therapeutic intervention. However, despite their predominant expression in immune cells such as microglia, there is limited information on P2X7 membrane expression and regulation during inflammation at the single-molecule level, necessitating new labeling approaches to visualize P2X7 in native cells. Here, we present **X7-uP**, an unbiased, affinity-guided P2X7 chemical labeling reagent that selectively and covalently biotinylates endogenous P2X7 in BV2 cells, a murine microglial cell line, allowing subsequent labeling with streptavidin-Alexa 647 tailored for super-resolution imaging. We uncovered a nanoscale microglial P2X7 redistribution mechanism where evenly spaced individual receptors in quiescent cells undergo upregulation and clustering in response to the pro-inflammatory agent lipopolysaccharide and ATP, leading to synergistic interleukin-1β release. Our method thus offers a new approach to revealing endogenous P2X7 expression at the single-molecule level.

## Introduction

ATP serves as the primary energy carrier in cells and also functions as an extracellular signaling molecule. Normally, extracellular ATP levels are typically low in healthy interstitial fluid, but they rise at

sites of stress or cellular injury, acting as a danger signal recognized as a damage-associated molecular pattern (DAMP). Extracellular ATP is detected by two families of purinergic receptors, metabotropic P2Y receptors and ionotropic P2X receptors (P2X1 to P2X7), with P2X7 specifically sensing abnormal ATP release in the mid-micromolar to low millimolar range (*Burnstock, 2004*; *Jacobson et al., 2020*; *Illes et al., 2021*). P2X7 is a transmembrane, non-selective cation channel ($Na^+$, $K^+$, $Ca^{2+}$) predominantly expressed in immune cells, such as macrophages and microglia (the resident macrophages of the central nervous system), where it plays a crucial role in inflammation and immunity (*North, 2002*; *Di Virgilio et al., 2017*). Upon ATP binding, P2X7 activation triggers $K^+$ efflux, promoting the assembly of the NLRP3 inflammasome and leading to the maturation and secretion of pro-inflammatory mediators, including interleukin-1β (IL-1β; *Ferrari et al., 1996*; *Ferrari et al., 1997*; *Monif et al., 2010*; *Bhattacharya and Biber, 2016*; *He et al., 2017*; *Illes, 2020*). P2X7 activation can cause cell death and is linked to the pathogenesis of several major disorders, including inflammatory pain (*Inoue and Tsuda, 2021*), autoimmune diseases (*Cao et al., 2019*), tumor progression (*Kan et al., 2019*; *Douguet et al., 2021*), psychiatric conditions (*Huang and Tan, 2021*), anxiety-like behavior (*Huang et al., 2024*), and neurodegenerative diseases (*Volonté et al., 2012*; *Martin et al., 2019*), making P2X7 a clinically relevant target.

At the cellular level, P2X7 is expressed at the plasma membrane and associates with lipid rafts, which are membrane microdomains known to play a key role in the onset of inflammation (*Garcia-Marcos et al., 2006*; *García-Marcos et al., 2006*; *Barth et al., 2007*; *Barth et al., 2008*; *Gonnord et al., 2009*; *Weinhold et al., 2010*). However, it is unknown how P2X7 is strictly distributed on the plasma membrane of cells and how its spatial and functional features intertwine in inflammatory conditions. Given the crucial role of P2X7 in inflammation and the growing interest in developing P2X7-targeted therapeutics, new approaches are needed to visualize P2X7 expression in immune cells at the single molecule level.

Numerous studies have investigated P2X7 expression in microglia using various experimental approaches, including functional assays (*Ferrari et al., 1996*; *Raouf et al., 2007*; *Janks et al., 2018*; *Trang et al., 2020*), radiolabeled ligand binding (*Lord et al., 2015*), immunofluorescence staining (*Choi et al., 2007*; *Monif et al., 2009*), electron microscopy (*Monif et al., 2016*), and genetic manipulations (*Kaczmarek-Hajek et al., 2018*). Although these studies confirmed the dominant expression of P2X7 in microglia, they provided only low-resolution cell imaging of P2X7, primarily due to the limitations of conventional light microscopy, which imposes a resolution restricted by the diffraction of light. Consequently, the detailed organization of P2X7 below ~250 nm remains unknown. This is a vital point, as recent data suggest that the spatial organization of cell membrane receptors may be a common regulatory mechanism for cellular signal transduction (*Manz and Groves, 2010*). However, it remains unclear whether such membrane organization similarly affects P2X7 signaling pathways.

Super-resolution microscopy techniques overcome this diffraction limit, enabling the visualization of fluorescently labeled membrane receptors at nanometric resolution (*Schermelleh et al., 2019*; *Choquet et al., 2021*). To the best of our knowledge, only two studies have reported super-resolved images of P2X7 (*Shrivastava et al., 2013*; *Gangadharan et al., 2015*). The first study found P2X7 in lipid microdomains of an osteoblastic cell line (*Gangadharan et al., 2015*), but the use of polyclonal antibodies to stain P2X7 raised questions about their selectivity (*Kaczmarek-Hájek et al., 2012*). The second study involved fusing P2X7 to the photo-convertible fluorescent Dendra2 protein to track its dynamic organization in hippocampal neurons (*Shrivastava et al., 2013*). While this study provided valuable insights into the nanometer-scale distribution of P2X7, it had three experimental biases. First, heterologous overexpression of P2X7-Dendra2 might not reflect native expression levels. Second, the C-terminal fusion of Dendra2 to P2X7 after the 'ballast' (*McCarthy et al., 2019*), a region that is critically involved in inflammation (*Di Virgilio et al., 2017*), could affect P2X7's pathogenicity. Third, there is growing evidence suggesting that neurons do not (or scarcely) express P2X7 (*Kaczmarek-Hajek et al., 2018*), questioning the physiological relevance of investigating P2X7 nanoscale organization in hippocampal neurons. Therefore, these issues prompted us to develop a new, genetic-free strategy able to decorate endogenous P2X7 in native cells with fluorophores suitable for super-resolution imaging.

Among existing methods for labeling endogenous proteins in living cells, the affinity-driven reaction strategy involving protein-ligand interaction is of particular interest (*Shiraiwa et al., 2020*). It relies on the formation of a covalent bond on the protein that is guided by the affinity of a ligand

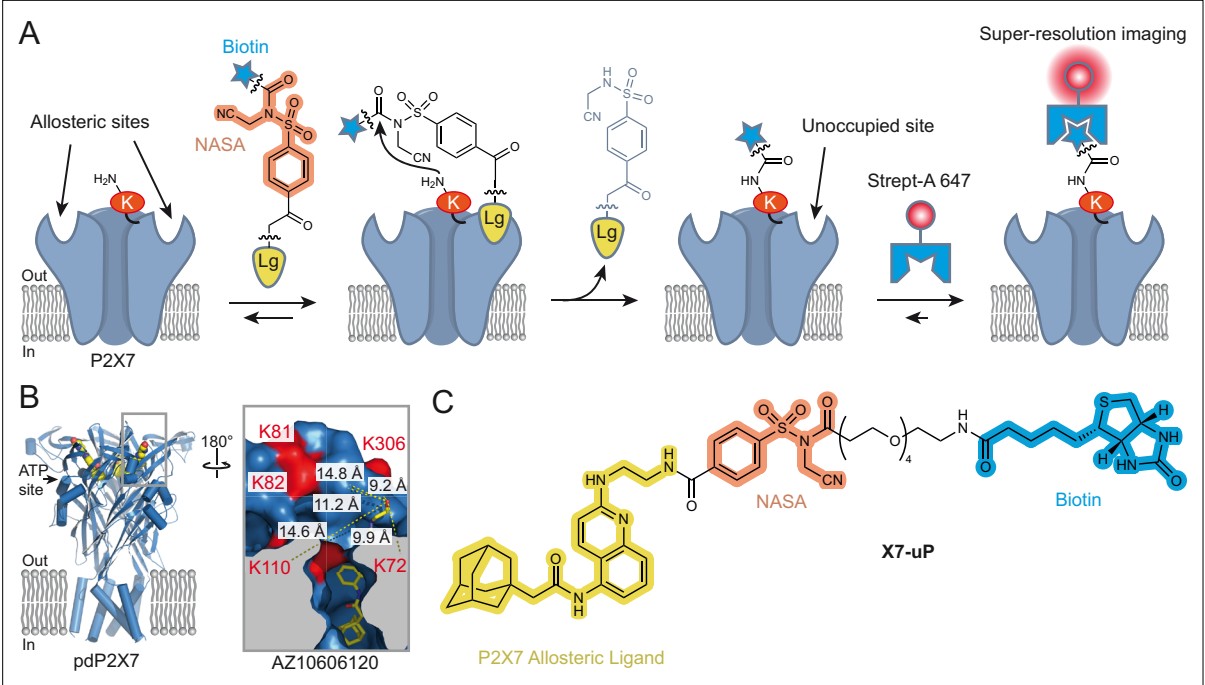

**Figure 1.** Affinity-guided labeling strategy for P2X7. (**A**) Schematic representation of the P2X7 labeling strategy using ligand-directed *N*-cyanomethyl NASA chemistry. A P2X7 ligand (Lg) binds to allosteric sites on the receptor, bringing the *N*-cyanomethyl NASA warhead into close proximity with endogenous lysine (K) residues. This spatial arrangement enables the covalent transfer of a biotin moiety via amide bond formation, resulting in highly specific biotinylation of P2X7. The biotin tag allows super-resolution imaging of nanoscale P2X7 localization using a Streptavidin-Alexa 647 probe (Strept-A 647). Notably, following biotin transfer, the ligand can dissociate from P2X7, leaving the allosteric sites unoccupied. (**B**) Crystal structure of panda P2X7 (pdP2X7) shown in ribbon representation, bound to AZ10606120 depicted as spheres (PDB:5U1W; *Karasawa and Kawate, 2016*). One of the three ATP-binding sites and the approximate location of the membrane are also indicated. Inset, enlarged view of the AZ10606120-binding pocket, rotated 180°. Distances (in Å) between the α-carbon of selected lysines and the hydroxyl group of AZ10606120 are displayed. Note that K300 is not visible in this view. (**C**) Chemical structure of **X7-uP**.

The online version of this article includes the following figure supplement(s) for figure 1:

**Figure supplement 1.** Mapping the X7-uP labeling site.

**Figure supplement 2.** Multi-step synthesis of X7-uP.

**Figure supplement 3.** ¹H NMR spectrum of compound 3.

**Figure supplement 4.** ¹H NMR spectrum of compound 5.

**Figure supplement 5.** ¹H NMR spectrum of compound 6.

**Figure supplement 6.** ¹H NMR spectrum of compound 7.

**Figure supplement 7.** NMR spectra of compound 8.

**Figure supplement 8.** NMR spectra of compound 10.

**Figure supplement 9.** NMR spectra of compound 11.

**Figure supplement 10.** NMR spectra of compound X7-uP.

bearing a reactive chemical moiety. The efficiency of the labeling reaction depends on the chemical function of the reactive species. The recently developed *N*-acyl-*N*-alkyl sulfonamide (NASA) chemistry offers a suitable platform that enables the covalent transfer, under native cell conditions, of functional ligands to nucleophilic amino acid residues (typically lysine residues) located in close proximity to the ligand-binding site (*Tamura et al., 2018*; *Figure 1A*). In addition, NASA electrophilicity can be finely tuned by introducing an electron-withdrawing moiety to the *N*-alkyl group, such as cyanomethyl, so that kinetic labeling can be considerably enhanced and carried out within minutes under physiological conditions (*Tamura et al., 2018*).

Here, we report **X7-uP**, a P2**X7-u**nbiased **P**urinergic labeling reagent that efficiently biotinylates native P2X7 lysine residues via a *N*-cyanomethyl NASA-based chemical reaction driven by the affinity of AZ10606120, a highly selective P2X7 allosteric ligand. Using streptavidin-Alexa Fluor 647 (Strept-A

647), we reveal the nanoscale distribution of P2X7 in BV2 cells at the single molecule level by direct stochastic optical-reconstruction microscopy (dSTORM). We show that P2X7 is homogenously dispersed on the plasma membrane as single receptors in quiescent (non-inflammatory condition) cells. However, upon exposure to the pro-inflammatory agent lipopolysaccharide (LPS), a pathogen-associated molecular pattern (PAMP) from Gram-negative bacteria, combined with ATP or its potent synthetic analog BzATP, P2X7 undergoes significant upregulation and clustering, resulting in the synergistic release of IL-1β. Our findings suggest that the dynamic clustering of P2X7 on the plasma membrane is a crucial mechanism underlying IL-1β secretion.

## Results

### Design and synthesis of X7-uP

**X7-uP** was rationally designed by leveraging available X-ray structures of the panda P2X7 receptor (pdP2X7) bound to non-competitive P2X7 antagonists (*Karasawa and Kawate, 2016*). These antagonists bind to an interfacial allosteric binding pocket in pdP2X7, distinct from ATP-binding sites, and accessible from the upper side of the trimeric receptor (*Karasawa and Kawate, 2016*). Due to the trimeric assembly of P2X subunits, three allosteric binding sites are present in P2X7 (*Figure 1B*). We focused on AZ10606120, a selective antagonist with high affinity for human (hP2X7) and rat P2X7 (rP2X7), with no effect on other P2X receptors (*Michel et al., 2007*; *Guile et al., 2009*; *Bhattacharya et al., 2013*; *Allsopp et al., 2017*). The X-ray structure revealed several lysine residues localized in proximity to AZ10606120 (<15 Å, measured from α-carbon of K72, K81, K82, K110, and K300, and the solvent-exposed hydroxyl of AZ10606120) (*Figure 1B* inset). Since most of these lysine residues are unique to P2X7 (*Figure 1—figure supplement 1A*), we hypothesized that NASA chemistry could selectively label P2X7. We modified AZ10606120 by replacing its solvent-exposed hydroxyethyl group with the reactive *N*-cyanomethyl NASA derivative, allowing a labeling reaction with the ε-amino group of nearby P2X7 lysine residues, forming a stable amide bond (*Figure 1A*). We used molecular docking to identify molecule **1** as a promising molecular scaffold built from AZ10606120 (*Figure 1—figure supplement 1B*). The distance separating the NASA electrophilic carbon of **1** and each α-carbon of nearby P2X7 residues was compatible with a proximity-driven reaction with rP2X7 (≤16.1 Å, *Figure 1—figure supplement 1C*), the receptor species used to experimentally validate the approach (see below). Thus, we designed and synthesized **X7-uP**, an extended version of **1**, containing an OEG (oligoethylene glycol) biotin tag (*Figure 1C*). **X7-uP** is expected to covalently transfer the biotin tag to P2X7, which, in turn, can serve as a versatile platform for subsequent cell surface labeling with commercially available, cell-impermeable biotin-binding protein-conjugated molecular probes, including Alexa 647, suitable for super-resolution microscopy (*Figure 1A*). **X7-uP** was synthesized as described in Materials and methods (*Figure 1—figure supplement 2*).

### X7-uP inhibits and biotinylates heterologously expressed rP2X7 in HEK293T cells

To address possible loss of affinity due to chemical modifications of the pharmacophore, we first assessed the ability of **X7-uP** to bind to P2X7 by measuring agonist-evoked current inhibition using whole-cell patch-clamp electrophysiology. In HEK293T cells transiently transfected with rP2X7, we recorded inward currents induced by eight successive applications of 10 µM 2'(3')-*O*-(4-benzoylbenzoyl) ATP (BzATP), a potent P2X7 agonist (*Figure 2A*). As expected, currents facilitated upon successive BzATP applications, a hallmark feature of P2X7 (*Surprenant et al., 1996*). Application of 1 µM of **X7-uP** for 10 s after washout of the fourth BzATP application did not induce inward currents but strongly inhibited currents induced by co-application for 2 s with BzATP (79.3 ± 5.6%, mean ± standard error of the mean (s.e.m.), n=4; control without inhibitors: –15.2 ± 4.4%, n=4, *Figure 2A and B*). Inhibition was reversible, as BzATP-evoked currents fully restored upon **X7-uP** washout (*Figure 2A*). Compared to the parental AZ10606120 compound, which produced nearly complete inhibition at 1 µM (99.7 ± 0.1%, n=4, *Figure 2A and B*), the strong but incomplete inhibition observed with **X7-uP** suggests that the chemical modification reduces its inhibitory potency at P2X7. To quantify this effect, we compared inhibition by 300 µM **X7-uP** with that produced by 10 nM AZ10606120, a concentration close to its reported IC$_{50}$ (*Bhattacharya et al., 2013*). Under these conditions, **X7-uP** produced a

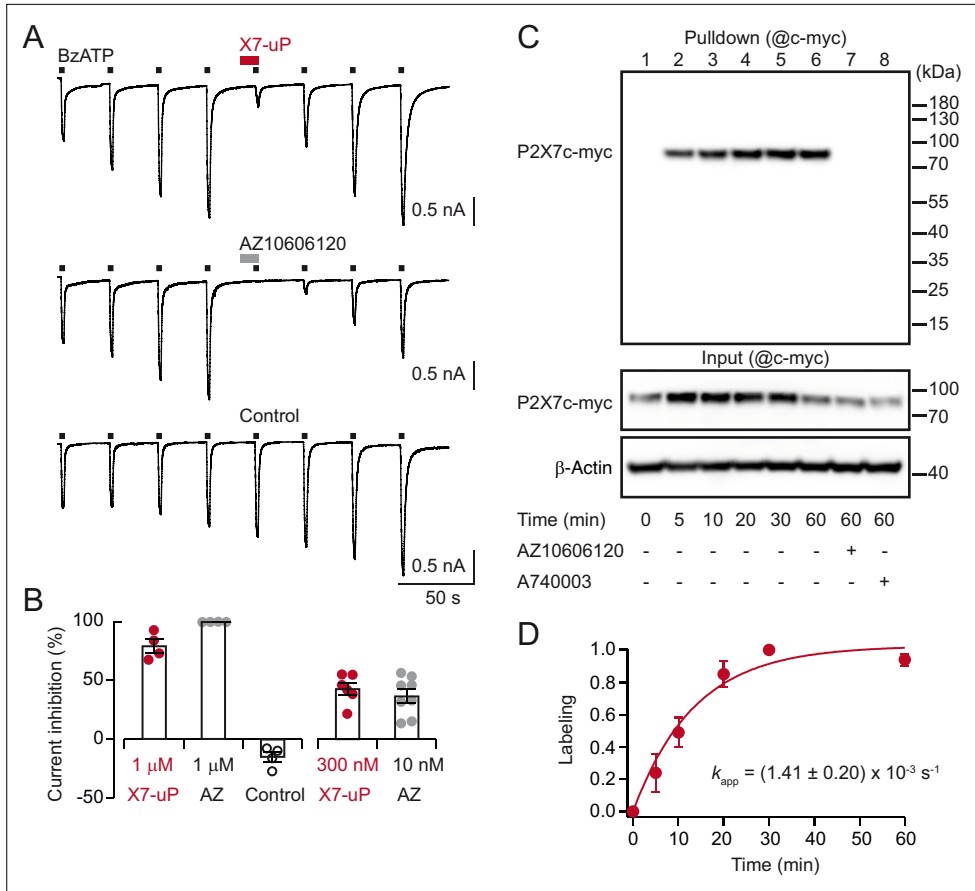

**Figure 2.** X7-uP is a potent P2X7 inhibitor that rapidly labels ectopically expressed P2X7 in HEK293T cells.
(**A**) Whole-cell currents evoked by 10 µM BzATP are reversibly inhibited by co-applying 1 µM **X7-uP** (upper trace) or 1 µM AZ10606120 (middle trace) in cells transiently transfected with rP2X7. Inhibitors were pre-applied alone for 10 s before 2 s of co-application. In the control (absence of inhibitors), BzATP-induced currents further increased, demonstrating current facilitation, which is an expected feature of P2X7 activation (***Surprenant et al., 1996***). (**B**) Summary of whole-cell inhibition at the indicated concentrations (n=4–8 cells for each condition). Bars represent mean ± s.e.m. (**C**) Western blot analysis of P2X7 labeling by **X7-uP**. Cells transiently transfected with P2X7c-myc were treated with 1 µM **X7-uP** for 0–60 minutes, in the absence or presence of 10 µM AZ10606120 or 10 µM A740003 (as indicated), followed by extensive washing. After cell lysis, biotinylated proteins were pulled down, separated on SDS-PAGE, and Western blotting was revealed using an anti-c-myc antibody (@c-myc). Molecular mass markers are shown on the right. Control of P2X7c-myc expression is presented in the corresponding input. β-Actin was used as a loading control. (**D**) Time course plot of P2X7 labeling with 1 µM **X7-uP**. Data (mean ± s.e.m., n=3 independent transfections) were fitted with ***Equation 1*** to determine the pseudo-first-order reaction rate $k_{app}$ (mean ± s.e.m.).

The online version of this article includes the following source data and figure supplement(s) for figure 2:

**Source data 1.** PDF file containing the original western blots used for ***Figure 2C***, indicating the relevant bands.

**Source data 2.** Original image files of western blots used for ***Figure 2C***.

**Source data 3.** Numerical values used to generate ***Figure 2B and D***.

**Figure supplement 1.** Specificity and kinetic analysis of P2X7 labeling by X7-uP.

**Figure supplement 1—source data 1.** PDF file containing the original western blots used for ***Figure 2—figure supplement 1A and B***, indicating the relevant bands.

**Figure supplement 1—source data 2.** Original image files of western blots used for ***Figure 2—figure supplement 1A and B***.

**Figure supplement 1—source data 3.** Numerical values used to generate ***Figure 2—figure supplement 1C and D***.

**Figure supplement 2.** Determination of X7-uP-induced P2X7 labeling yields in HEK293T and BV2 cells.

*Figure 2 continued on next page*

*Figure 2 continued*

**Figure supplement 2—source data 1.** PDF file containing the original western blots used for *Figure 2—figure supplement 2A*, indicating the relevant bands.

**Figure supplement 2—source data 2.** Original image files of western blots used for *Figure 2—figure supplement 2A*.

**Figure supplement 2—source data 3.** Numerical values used to generate *Figure 2—figure supplement 2B*.

similar level of inhibition (*Figure 2B*), indicating that although its potency is markedly reduced relative to AZ10606120, it nonetheless remains an effective P2X7 probe at 1 µM.

To test labeling efficiency, HEK293T cells were transiently transfected with c-myc tagged rP2X7 (P2X7c-myc) and incubated at 20 °C with 1 µM **X7-uP** for 60 min. **X7-uP** labeling was performed under physiological conditions (1 µM for 10 min at 37 °C in FBS-free DMEM, see Materials and methods), followed by extensive washing to remove excess **X7-uP**. After cell lysis, biotinylated proteins were pulled down with NeutrAvidin Agarose, separated on SDS-PAGE, and revealed with an anti-c-myc antibody in western blot analysis. A strong and unique band was detected in the western blot at the expected apparent molecular mass of monomeric P2X7c-myc (~75–80 kDa; *Figure 2C*). However, this band disappeared upon co-incubation with AZ10606120 (10 µM), or in the absence of **X7-uP** (lane 1), while robust signals corresponding to P2X7c-myc and β-actin expression were still detected in the corresponding input controls (*Figure 2C*). Similarly, no signal was detected in non-transfected cells treated with **X7-uP** (*Figure 2—figure supplement 1A*). To further demonstrate the pharmacological specificity of P2X7 labeling by **X7-uP**, we used another non-competitive P2X7 antagonist, A740003 (10 µM), which targets the same allosteric binding cavity as AZ10606120 in pdP2X7 but is chemically different from AZ10606120 (*Karasawa and Kawate, 2016*). As expected, no labeling was detected when A740003 was co-incubated with **X7-uP** (*Figure 2C*). Taken together, these data strongly suggest that **X7-uP** biotinylates P2X7 residues by targeting the unique P2X7 allosteric binding site.

We next conducted a kinetic analysis of P2X7 labeling by incubating cells with **X7-uP** at different concentrations and monitoring labeling through quantitative Western blotting analysis. The time course of P2X7 labeling at 1 µM of **X7-uP** is shown in *Figure 2C and D*, with labeled bands detected as early as 5 minutes of incubation time. The data were fitted to *Equation 1*, derived according to a kinetic model in which an irreversible chemical reaction follows a reversible ligand binding reaction in a large excess of ligand (see Methods) (*Tamura et al., 2018*), providing the pseudo-first-order reaction rate ($k_{app}$) of the labeling reaction. Kinetics of labeling carried out at different **X7-uP** concentrations allowed us to plot $k_{app}$ values against **X7-uP** concentrations (*Figure 2—figure supplement 1B–D*). The data were fitted to *Equation 2*, providing the labeling rate constant $k_L$ (0.011±0.003 s$^{-1}$, mean of triplicate ± standard deviation (s.d.)), the dissociation constant $K_d$ (7.3±2.7 µM, mean ± s.d.), and the second-order rate constant ($k_L / K_d = 1.5 \times 10^3$ M$^{-1}$s$^{-1}$). The second-order rate constant value was comparable to those of previously described labeling reagents based on NASA chemistry (*Tamura et al., 2018*).

Next, we determined **X7-uP** labeling yield by quantifying the amount of P2X7c-myc in the supernatant after pulling down biotinylated P2X7c-myc (*Figure 2—figure supplement 2A*). This amount, corresponding to unbiotinylated P2X7c-myc, was then compared to the maximal amount of unbiotinylated P2X7c-myc carried out in the absence of **X7-uP**, thus providing labeling yield. Quantitative Western blotting analysis revealed that 65 ± 3% (mean of triplicate ± s.e.m.) of P2X7c-myc were labeled by 1 µM of **X7-uP** for 60 min in HEK293T cells (*Figure 2—figure supplement 2B*).

## X7-uP labeling is selective for P2X7

Having established the efficacy of P2X7 labeling by Western blot, we next assessed the selectivity of **X7-uP** towards the P2X family. We performed **X7-uP** labeling in HEK293T cells transiently transfected with various fluorescently tagged rat P2X subunits (P2X1 to P2X7). The biotinylated cells were then visualized using Strept-A 647 labeling through confocal microscopy. These P2X subtypes are closely related to P2X7 but are not expected to exhibit the AZ10606120 allosteric site.

As expected, confocal microscopy revealed a strong Alexa 647 fluorescence signal in the periphery of cells expressing only P2X7 tagged with the monomeric Scarlet (mScarlet) fluorescent protein (*Figure 3A*). In contrast, no Alexa 647 signal was detected in cells transiently transfected with other P2X subunits tagged with GFP, although a strong GFP signal was observed for all constructs, confirming

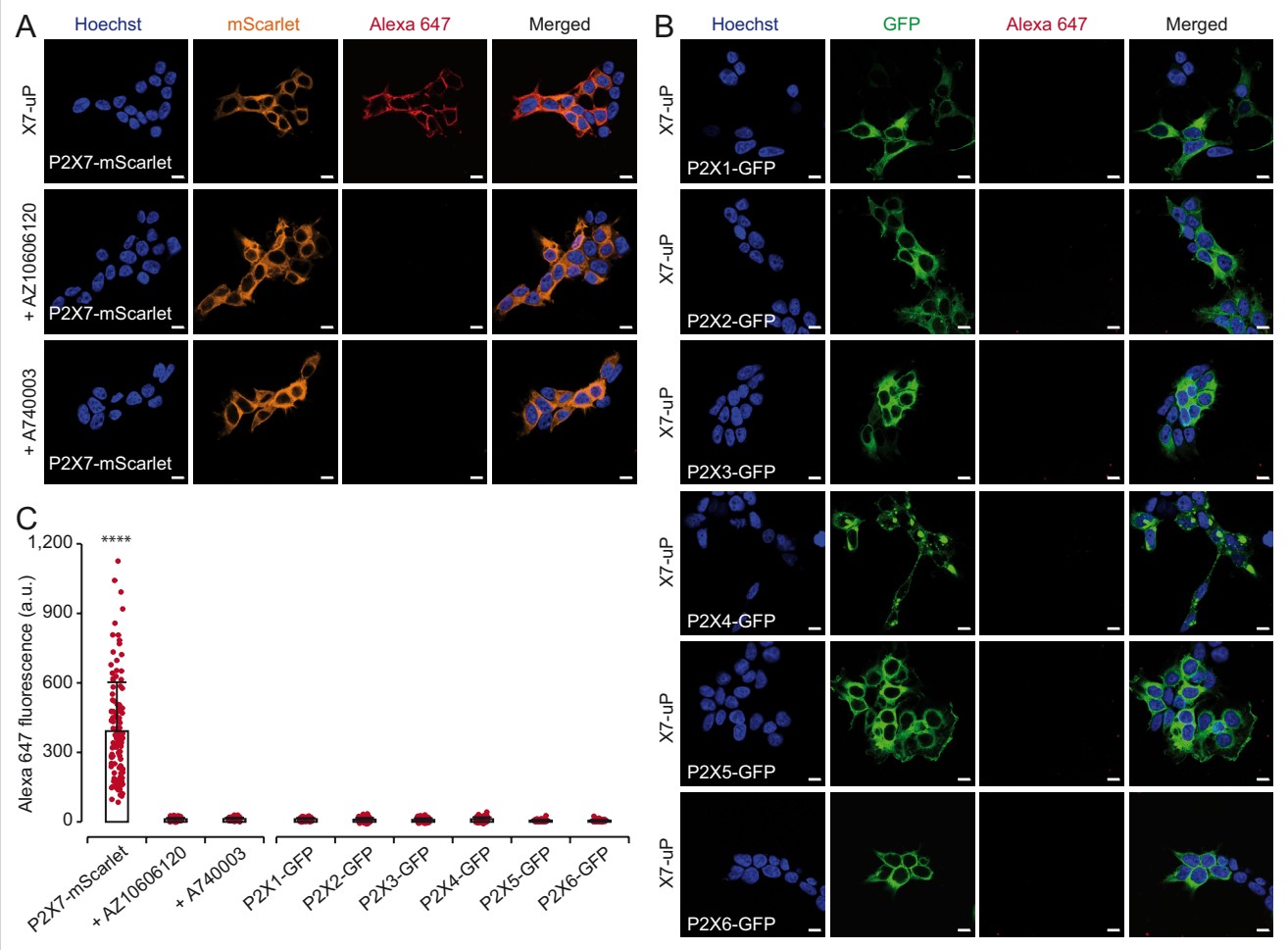

**Figure 3.** X7-uP labeling is highly selective for P2X7. (**A–B**) Confocal images of HEK293T cells transiently transfected with either P2X7-mScarlet (**A**) or various P2X subunits tagged with GFP (P2X1-GFP, P2X2-GFP, P2X3-GFP, P2X4-GFP, P2X5-GFP, or P2X6-GFP) (**B**) were labeled with **X7-uP** and revealed using Strept-A 647 (red) in FBS-free DMEM. Labeling was performed in the presence of 10 µM AZ10606120 or 10 µM A740003 (**A**). Nuclei were stained with Hoechst (blue). For clarity, mScarlet and GFP signals are displayed in orange and green, respectively. Scale bars, 10 µm. (**C**) Quantification of Alexa 647 fluorescence. Bars represent mean ± standard deviation (s.d.) (n=75–129 cells, *t*-test comparisons to P2X7-mScarlet, ****p<0.0001).

The online version of this article includes the following source data, source code, and figure supplement(s) for figure 3:

**Source code 1.** RStudio code to generate the histogram of *Figure 3C*.

**Source data 1.** Data set used to generate *Figure 3C*.

**Figure supplement 1.** Cell-surface labeling assay of HEK293T cells transfected with various GFP-tagged P2X subunits.

**Figure supplement 1—source data 1.** PDF file containing the original western blots used for *Figure 3—figure supplement 1A and B*, indicating the relevant bands.

**Figure supplement 1—source data 2.** Original image files of western blots used for *Figure 3—figure supplement 1A and B*.

**Figure supplement 1—source data 3.** Numerical values used to generate *Figure 3—figure supplement 1C and D*.

**Figure supplement 2.** dSTORM images of HEK293T cells transfected with P2X7-mScarlet.

correct expression (*Figure 3B and C*). Using a commercial cell-surface protein labeling assay, we further confirmed that tagged P2X subunits were expressed at the cell surface—except P2X6, as expected—supporting proper membrane trafficking (*Figure 3—figure supplement 1*). Additionally, co-incubation with AZ10606120 (10 µM) or A740003 (10 µM) during **X7-uP** labeling abolished the Alexa 647 signal but did not affect mScarlet signal, further confirming the specificity of P2X7 labeling (*Figure 3A and C*). These data demonstrate the high selectivity of **X7-uP** for P2X7 labeling.

To further validate our method for single-molecule localization microscopy, we employed dSTORM to visualize the fluorescence emission of individual Alexa 647 fluorophores bound to P2X7 expressed

on the plasma membrane of HEK293T cells. Our results confirmed the expression of P2X7 at the cell surface, consistent with confocal microscopy data (*Figure 3—figure supplement 2A*). Importantly, no emission was detected when labeling was conducted in the presence of AZ10606120 (10 μM), or when Strept-A 647 was added to cells not treated with **X7-uP** (*Figure 3—figure supplement 2B* and C), indicating that the observed emission in dSTORM specifically originated from P2X7.

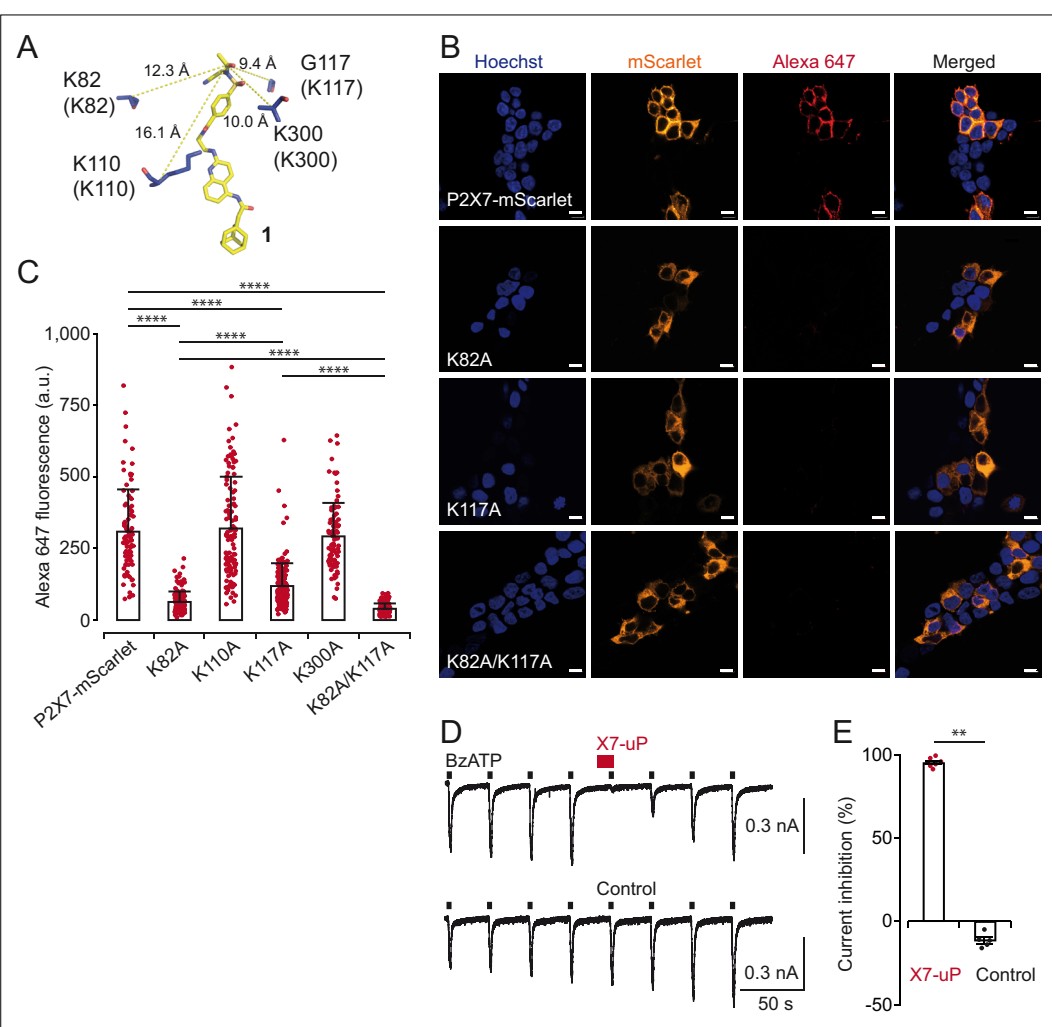

**Figure 4.** X7-uP labels K82 and K117 in rat P2X7. (**A**) Molecular docking of **1** (same pose as shown in *Figure 1—figure supplement 1C*) in pdP2X7, showing distances (in Å) between the reactive carbonyl of **1** (stick representation) and selected α-carbons of nearby residues (blue). Residues shown in parentheses correspond to equivalent rP2X7 residues. (**B**) Confocal images of HEK293T cells transiently transfected with different P2X7 constructs: P2X7-mScarlet, K82A, K117A, and K82A/K117A. Refer to the legend of *Figure 3* for additional details. Scale bars, 10 μm. (**C**) Quantification of Alexa 647 fluorescence. Bars represent mean ± s.d. (n=90–190 cells, *t*-test comparisons to indicated conditions, ****p<0.0001). (**D**) Whole-cell currents evoked by 10 μM BzATP are reversibly inhibited by co-application of 0.5 μM **X7-uP** (upper trace) to BzATP in a cell transiently transfected with the double mutant K82A/K117A. The control (absence of **X7-uP**) is shown in the bottom trace. (**E**) Summary of whole-cell inhibition for K82A/K117A (n=7 cells for **X7-uP** and 5 cells for control). Bars represent mean ± s.e.m.; Mann-Whitney test (**p<0.005).

The online version of this article includes the following source data and figure supplement(s) for figure 4:

**Source data 1.** Data set and numerical values used to generate *Figure 4C and E*.

**Figure supplement 1.** Additional data related to *Figure 4B*.

## X7-uP labels K82 and K117 in rP2X7

We next identified the labeling site of **X7-uP**. Since it has been shown that the NASA reagent preferentially reacts with the ε-amino group of lysine side chains (*Tamura et al., 2018*), we individually mutated K82, K110, K117, and K300 into alanine, an amino acid residue that cannot react with NASA, in the P2X7-mScarlet background. These residues were selected based on the predicted proximity of their equivalent positions in pdP2X7 to the electrophilic NASA group of **1** in pdP2X7 (*Figure 4A*, *Figure 1—figure supplement 1A*). We expressed these mutants individually in HEK293T cells and monitored **X7-uP** labeling by quantifying Alexa 647 signals through confocal microscopy. While K110A and K300A mutants did not affect **X7-uP** labeling, where Alexa 647 fluorescence quantification showed comparable levels to that of P2X7-mScarlet (*Figure 4C*, *Figure 4—figure supplement 1*), a significant reduction in Alexa 647 fluorescence was observed for K82A (79.6 ± 1.0 %, n=154 cells, p<0.0001, *t*-test to the respective control P2X7-mScarlet) and K117A (61.5 ± 2.1%, n=155 cells, p<0.0001). Importantly, a strong mScarlet signal was detected for all mutants, indicating successful expression of mutants in cells (*Figure 4B*, *Figure 4—figure supplement 1*). Alexa 647 fluorescence reduction at K82A was significantly different from that at K117A (p<0.0001, *t*-test), suggesting a larger contribution of K82 to **X7-uP** labeling. The double mutant K82A/K117A further reduced the signal by 87.3 ± 0.4% (n=199 cells, p<0.0001, *t*-test to respective control P2X7-mScarlet, K82A, or K117A), without affecting P2X7 expression (*Figure 4B and C*). To exclude the possibility that introduced mutations affected **X7-uP** binding, we further demonstrated that the double K82A/K117A mutant did not affect the ability of **X7-uP** to inhibit BzATP-evoked currents (*Figure 4D and E*). These data support the hypothesis that both K82 and K117 are the sites of **X7-uP** labeling in rP2X7.

## X7-uP reveals uniform membrane distribution of P2X7 in quiescent BV2 cells

Our data demonstrate the high selectivity of **X7-uP** for rP2X7 in HEK293T cells. We next investigated whether its application extends to native P2X7-expressing cells to reveal endogenous P2X7 expression. We chose BV2 cells, a cultured murine microglial cell line in which functional P2X7 expression has been previously documented (*Raouf et al., 2007*; *Trang et al., 2020*). In addition, activation of BV2 cells to mimic microglial inflammation can be effectively stimulated in vitro using both LPS and ATP (*Huang et al., 2024*). Although K117 is not conserved in mouse P2X7, the highly conserved K82 should ensure the successful labeling of mouse P2X7 with **X7-uP** (*Figure 1—figure supplement 1A*).

We first confirmed by ELISA assay that, under untreated conditions, BV2 cells are quiescent as they do not release IL-1β (*Figure 5A and B*). We also confirmed P2X7 expression by Western blot and confocal microscopy following 1 µM **X7-uP** labeling for 10 min in FBS-free DMEM (*Figure 5—figure supplement 1*). As observed for HEK293T cells, a strong and unique band was detected using a P2X7 antibody only in the presence of **X7-uP** in the pulldown Western blot, at the expected apparent molecular mass of monomeric mouse P2X7 (~68 kDa, unglycosylated form; see lanes 3 and 4 in *Figure 5—figure supplement 1A*). This band disappeared upon co-incubation with AZ10606120 or A740003, while controls in the supernatant consistently showed endogenous P2X7 expression in all lanes, though non-specific bands were also observed, presumably due to the limited specificity of commercially available P2X7 antibodies (*Figure 5—figure supplement 1B*). Quantitative Western blotting analysis revealed that 80 ± 1% (mean of triplicate ± s.e.m.) of total P2X7 were labeled in BV2 cells (*Figure 2—figure supplement 2A* and B). Confocal microscopy revealed a strong Alexa 647 fluorescence signal in the periphery of BV2 cells that was reduced upon co-incubation with AZ10606120 or A740003 (*Figure 5—figure supplement 1C* and D). Control with Strept-A 647 alone showed an extremely low fluorescence background level (*Figure 5—figure supplement 1C* and D), demonstrating high **X7-uP** labeling specificity in BV2 cells.

To assess how BV2 activation affects P2X7 molecular organization, we employed dSTORM to track single fluorophores tagged to endogenous receptors. In this super-resolution technique, each fluorophore blinks several times before becoming silent due to bleaching or reaching a stable dark state. As a result, clusters of detections could result either from multiple blinks of a single receptor or represent the sum of multiple fluorophore emissions from clustered receptors. To quantify nanoscale object properties, we used Metamorph to obtain pixelized global images for global quantification (*Figure 5—figure supplement 2D*), and SR-Tesseler, an open-source segmentation framework based

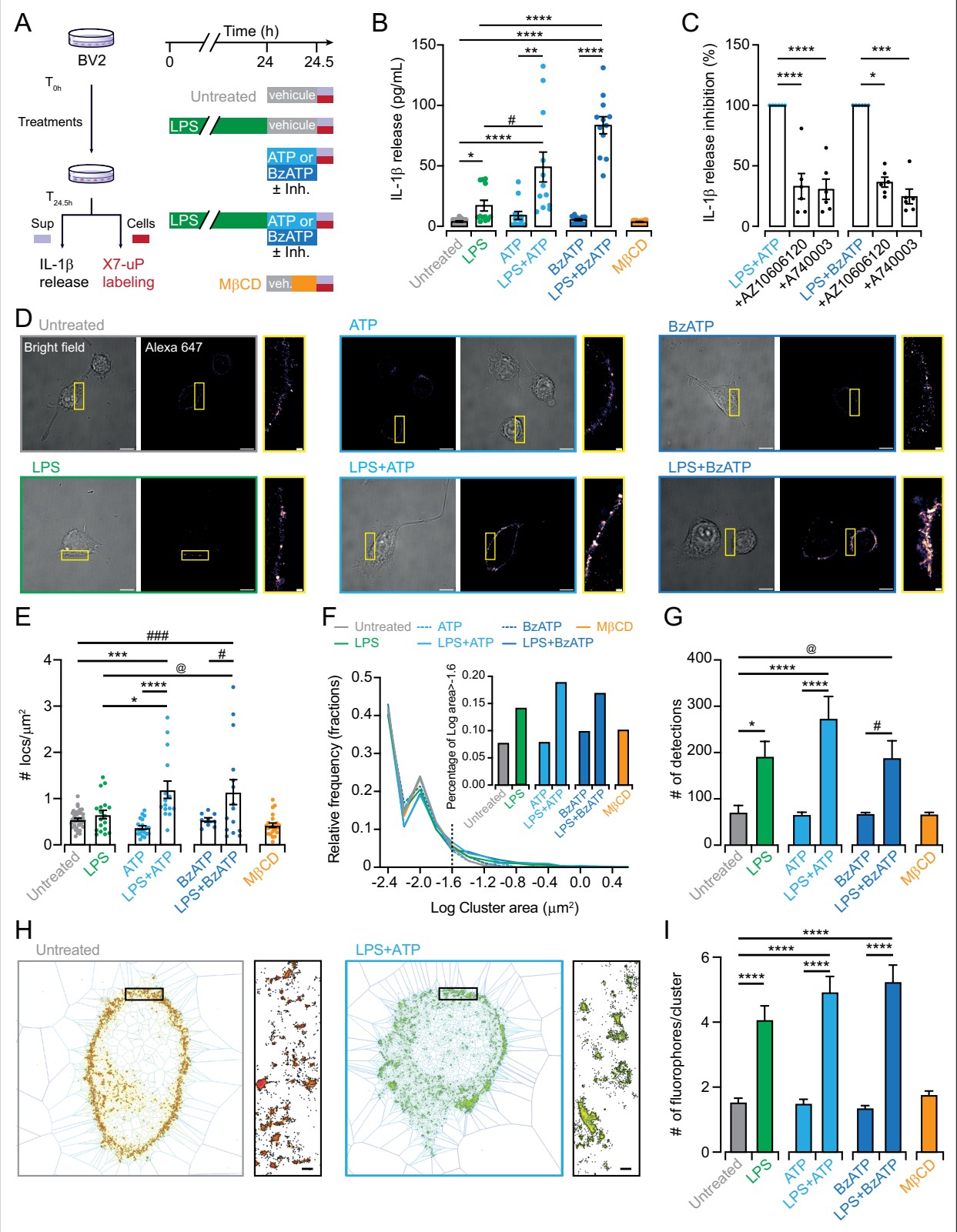

**Figure 5.** dSTORM data revealed nanoscale P2X7 plasma membrane localization in BV2 cells. (**A**) Cartoon and experimental timeline of BV2 cell treatments. IL-1β release was assessed in the supernatant (sup), and the same cells were labeled with 1 μM **X7-uP** after extensive washout. (**B**) Quantification of IL-1β release by ELISA following the indicated treatments: LPS (1 μg/mL for 24 hr), ATP (1 mM for 30 min), BzATP (300 μM for 30 min), and MβCD (15 mM for 15 min). Bars represent mean ± s.e.m. (n=12 samples from three independent experiments). Data were compared using

*Figure 5 continued on next page*

*Figure 5 continued*

Kruskal-Wallis followed by Dunn's multiple comparisons (*p=0.0208, #p=0.0362, **p=0.0014, ****p<0.0001). (C) Normalized quantification of IL-1β release induced by LPS +ATP or LPS +BzATP in the presence of P2X7 inhibitors AZ10606120 or A740003. Bars represent mean ± s.e.m. (n=6 samples from six independent experiments). One-way ANOVA with Dunnett's multiple comparisons to control condition for ATP data (****p<0.0001). Kruskal-Wallis followed by Dunn's multiple comparisons to control condition for BzATP data (*p=0.0414, *** p=0.0006). (D) Bright-field and dSTORM images of **X7-uP**-labeled BV2 cells revealed with Strept-A 647 corresponding to experiments shown in panel B. Scale bars, 10 μm. Insets: Magnified dSTORM images. Scale bars, 1 μm. (E) Quantification of single P2X7 localization density. Bars represent mean ± s.e.m. (each data point represents a cell, n=3 independent experiments). One-way ANOVA with Tukey's multiple comparisons (*p<0.019, #p<0.0194, @p<0.0477, ***p<0.0002, ###p<0.0008, ****p<0.0001). (F) Relative frequency of cluster size. Inset: percentage of clusters larger than 0.025 mm$^2$. (G) Number of detections per cluster. Bars represent mean ± s.e.m. One-way ANOVA with Tukey's multiple comparisons (*p<0.0197, #p<0.0277, @p<0.0389, ****p<0.0001). (H) Images showing tessellation analysis of cells treated either with LPS +ATP or left untreated. Inset: magnification. Scale bars, 200 nm. (I) Number of fluorophores per cluster. Bars represent mean ± s.e.m. One-way ANOVA with Tukey's multiple comparisons (****p<0.0001).

The online version of this article includes the following video, source data, and figure supplement(s) for figure 5:

**Source data 1.** Numerical values used to generate *Figure 5B, C, E, F, G and I*.

**Figure supplement 1.** X7-uP-induced P2X7 labeling in BV2 cells.

**Figure supplement 1—source data 1.** PDF file containing the original western blots used for *Figure 5—figure supplement 1A and B*, indicating the relevant bands.

**Figure supplement 1—source data 2.** Original image files of western blots used for *Figure 5—figure supplement 1A and B*.

**Figure supplement 1—source data 3.** Data set used to generate *Figure 5—figure supplement 1D*.

**Figure supplement 2.** dSTORM analysis of X7-uP-induced P2X7 labeling in BV2 cells.

**Figure supplement 2—source data 1.** Numerical values used to generate *Figure 5—figure supplement 2B, F and G*.

**Figure supplement 3.** Blink detection features.

**Figure supplement 3—source data 1.** Numerical values used to generate *Figure 5—figure supplement 3A and B*.

**Figure supplement 4.** Confocal data of X7-uP-induced P2X7 labeling in BV2 cells.

**Figure supplement 4—source data 1.** Numerical values used to generate *Figure 5—figure supplement 4B*.

**Figure 5—video 1.** dSTORM movie of untreated BV2 cells.

https://elifesciences.org/articles/106096/figures#fig5video1

**Figure 5—video 2.** dSTORM movie of LPS +ATP-treated BV2 cells.

https://elifesciences.org/articles/106096/figures#fig5video2

on Voronoï tessellation from localized molecule coordinates (*Levet et al., 2015*), to analyze cluster properties (*Figure 5H and I*).

In the control untreated condition (quiescent cells), dSTORM revealed punctated images, while no detections occurred without **X7-uP**, further confirming high labeling specificity in BV2 cells (*Figure 5D*, *Figure 5—figure supplement 2A and B*, *Figure 5—figure supplement 3A* and *Figure 5—video 1*). Detections are organized into clusters of 44 nm in size (median, 25% percentile 18 nm, 75% percentile 80 nm), with on average 70 ± 15 (mean ± s.e.m.) detections per cluster (n=4 cells, *Figure 5G*). Distribution analysis suggests that detections of fluorescent streptavidin conjugates are originating from one single receptor. On average, every P2X7 receptor is separated from its neighbor by a mean inter-cluster distance of 292 ± 5 nm (*Figure 5—figure supplement 2G*). These data suggest that P2X7 is homogenously dispersed on the cell surface of quiescent BV2 cells.

Since P2X7 has been suggested to be associated with lipid rafts (*Garcia-Marcos et al., 2006*; *García-Marcos et al., 2006*; *Barth et al., 2007*; *Barth et al., 2008*; *Gonnord et al., 2009*; *Weinhold et al., 2010*), we pre-incubated cells with 15 mM methyl-β-cyclodextrin (MβCD), a lipid-raft disrupting agent, for 15 min, and labeled P2X7 with **X7-uP** after extensive washout to remove remaining MβCD (*Figure 5A*). MβCD treatment had no effect on IL-1β release, the number of detections per cluster, cluster size, or inter-cluster distance (*Figure 5B, F and G*, *Figure 5—figure supplement 2C and G* and G). We interpret these results to suggest that disrupting lipid rafts does not alter P2X7 distribution in quiescent BV2 cells.

## X7-uP reveals P2X7 clustering and upregulation in activated BV2 cells

We analyzed changes in cluster properties following BV2 activation by LPS and P2X7 agonists. To reliably correlate P2X7 spatial distribution with the inflammatory response, we labeled the same BV2

cells used for IL-1β release assays with **X7-uP** (*Figure 5A*). Previous studies have proposed a two-signal, synergistic model for IL-1β release in microglia and other cell types (*Ferrari et al., 1996*; *Perregaux et al., 2000*; *Ferrari et al., 2006*; *Di Virgilio et al., 2017*; *He et al., 2017*; *Swanson et al., 2019*), where the first signal involves LPS priming, leading to the intracellular accumulation of pro-IL-1β, while the second involves ATP-dependent P2X7 stimulation, promoting caspase-1 activation and the release of mature IL-1β. We confirmed this model by observing robust IL-1β release when BV2 cells were exposed to 1 µg/mL LPS for 24 hr, followed by 1 mM ATP (or 300 µM BzATP) for 30 min (LPS +ATP or LPS +BzATP; *Figure 5B*). No IL-1β release was detected when ATP or BzATP was applied alone, while LPS alone induced significant release. However, this release was significantly less than that induced by co-treatments, representing only 35% of the release observed with LPS +ATP and 21% with LPS +BzATP (*Figure 5B*), confirming the synergistic effect. Co-incubation with AZ10606120 or A740003 during agonist exposure reduced IL-1β release to levels observed with LPS alone, demonstrating the role of P2X7 in IL-1β release (*Figure 5A and C*).

dSTORM images following treatments are shown in *Figure 5D*. Global single-molecule labeling on the cell surface was significantly more pronounced in cells treated with LPS +ATP or LPS +BzATP, and to a lesser extent, though not significantly, in those treated with LPS alone compared to untreated control (*Figure 5*, *Figure 5—figure supplement 3B*, and *Figure 5—video 2*). Similar results were obtained using confocal microscopy, although the effect of LPS alone was significant (*Figure 5—figure supplement 4A and B* and B). Neither ATP nor BzATP application alone affected the number of detections per cluster, cluster size, or inter-cluster distance. At the cluster level, both LPS and LPS +ATP or LPS +BzATP treatments increased the number of detections per cluster (*Figure 5E-G*, *Figure 5—figure supplement 2D-E*), with no change in inter-cluster distance or average intensity per sub-pixel (*Figure 5—figure supplement 2F* and G), indicating no increase in detection density per pixel. On pixelized images, larger clusters (area greater than 0.025 mm$^2$, insert *Figure 5F*) emerged following LPS treatment, with or without agonists. Tessellation analysis revealed that the number of fluorophores within clusters significantly increased from ~1.5 in untreated cells to 4.1 ± 0.4 for LPS, 4.9 ± 0.5 for LPS +ATP, and 5.2 ± 0.5 for LPS +BzATP (*Figure 5H and I*). These data demonstrate that pro-inflammatory conditions induce P2X7 clustering by upregulating the average number of P2X7 receptors within clusters, increasing it from one to three, and favoring the formation of larger clusters (*Figure 6*). The strong P2X7 clustering induced by the combination of LPS and P2X7 agonists mirrors IL-1β secretion.

## Discussion

We present a rapid and efficient method for the selective labeling of P2X7 in native cells. This approach allowed us to uncover a nanoscale redistribution mechanism of individual endogenous P2X7 receptors on the plasma membrane of activated BV2 microglial cells, leading to the synergistic release of the pro-inflammatory cytokine IL-1β.

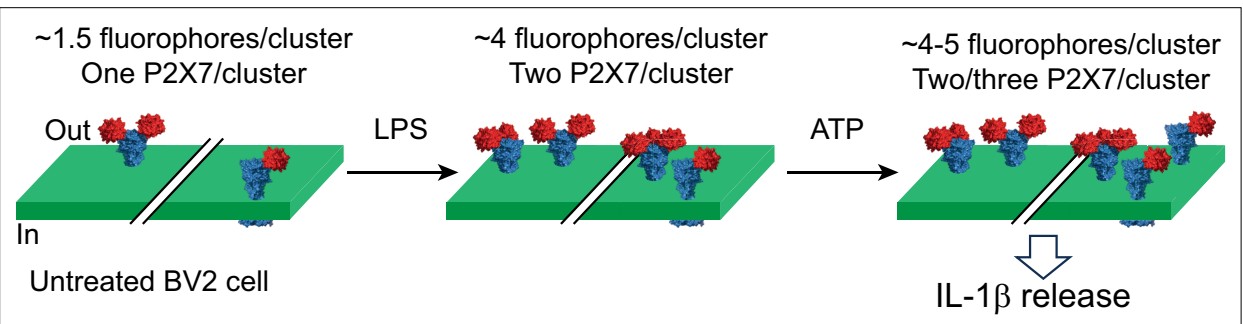

**Figure 6.** Nanoscale redistribution of individual P2X7 receptors in microglia under pro-inflammatory conditions at the plasma membrane. The cartoon illustrates two distinct clusters of P2X7 receptors (blue), each adorned with one, two, or three fluorescently tagged tetrameric biotin-bound streptavidin (red). In untreated cells, each cluster contains an average of 1.5 fluorophores per P2X7 receptor. Treatment with LPS and ATP promotes P2X7 clustering by increasing the average number of fluorophores per cluster to between 4 and 5, resulting in an increased number of P2X7 receptors per cluster, from one to three. This redistribution synergistically triggers IL-1β release.

Our strategy employs an affinity-guided approach, through which we developed **X7-uP**, a selective P2X7 biotinylating reagent that covalently attaches a biotin tag to lysine residues on native P2X7. In rat P2X7 (rP2X7), we identified two lysine residues as labeling sites: K82, a highly conserved residue unique to P2X7, and K117, a residue found only in rP2X7. We found that K82 is the primary labeling site in rP2X7, and given that K117 is absent in mouse P2X7, it is likely that K82 is the only labeled site in BV2-expressing P2X7.

We demonstrate the effectiveness of **X7-uP** in both heterologous and native expression systems. Compared to existing methods, **X7-uP** offers several advantages. First, it enables rapid labeling of P2X7 within 10 min at low **X7-uP** concentrations (1 µM) in a physiological-like buffer (e.g. FBS-free DMEM). This efficiency is likely due to the use of *N*-cyanomethyl NASA chemistry coupled with the high-affinity ligand AZ10606120. Second, **X7-uP** is highly selective for P2X7: no labeling is observed in cells expressing other P2X subtypes or in the presence of P2X7-specific antagonists. This selectivity arises from AZ10606120's binding to a unique allosteric site on P2X7 (*Karasawa and Kawate, 2017*), making **X7-uP** particularly suitable for detecting P2X7 in native tissues. This level of specificity outperforms that of commercially available P2X7 antibodies, which often suffer from poor specificity and are generally unreliable for detecting endogenous receptors (*Sim et al., 2004*; *Anderson and Nedergaard, 2006*; *Kaczmarek-Hájek et al., 2012*). While recent nanobody-based approaches have improved P2X7 specificity (*Kaczmarek-Hajek et al., 2018*; *Sierra-Marquez et al., 2024*), our approach is mechanistically distinct and complementary to nanobody-based methods. Third, **X7-uP** achieves covalent biotinylation of P2X7 lysine residues via stable amide bond formation, ensuring that the biotin moiety remains permanently attached, an advantage not afforded by reversible binding strategies. Fourth, **X7-uP** provides a versatile platform to deliver various probes to P2X7, including those for pull-down assays and super-resolution imaging. The strong biotin-streptavidin (or neutravidin) interaction ensures specificity and stability, two crucial criteria for biochemical assays and cell imaging.

Our results not only confirm P2X7 expression in BV2 microglial cells, as previously reported (*Ferrari et al., 1996*; *Choi et al., 2007*; *Raouf et al., 2007*; *Monif et al., 2009*; *Lord et al., 2015*; *Monif et al., 2016*; *Janks et al., 2018*; *Kaczmarek-Hajek et al., 2018*; *Trang et al., 2020*), but also reveal its nanoscale localization at the cell surface using dSTORM. In quiescent cells, P2X7 is evenly distributed across the plasma membrane, likely as individual receptors with an average nearest-neighbor distance of ~300 nm, indicating the absence of clustering. To test whether P2X7 associates with cholesterol-enriched lipid rafts, we pre-treated cells with MβCD, a cholesterol-depleting agent that perturbs lipid rafts. This treatment did not affect P2X7 distribution, although we have previously shown in HEK293T cells that MβCD increases P2X7 single-channel mean open time without altering unitary conductance (*Dunning et al., 2021*). These findings suggest that in quiescent BV2 microglial cells, perturbing lipid rafts does not alter the uniform distribution of individual P2X7 receptors, which likely reside within cholesterol-enriched nanodomains.

Upon BV2 activation, we observed significant nanoscale reorganization of P2X7. Both LPS and ATP (or BzATP) trigger P2X7 upregulation and clustering, increasing the overall number of surface receptors and the number of receptors per cluster, from one to three (*Figure 6*). By labeling BV2 cells with **X7-uP** shortly after IL-1β release, we were able to correlate the nanoscale distribution of P2X7 with the functional state of BV2 cells, consistent with the two-signal, synergistic model for IL-1β secretion observed in microglia and other cell types (*Ferrari et al., 1996*; *Perregaux et al., 2000*; *Ferrari et al., 2006*; *Di Virgilio et al., 2017*; *He et al., 2017*; *Swanson et al., 2019*). In this model, LPS priming leads to intracellular accumulation of pro-IL-1β, while ATP stimulation activates P2X7, triggering NLRP3 inflammasome activation and the subsequent release of mature IL-1β.

What is the mechanism underlying P2X7 upregulation that leads to an overall increase in surface receptors—does it result from the lateral diffusion of previously masked receptors already present at the plasma membrane, or from the insertion of newly synthesized receptors from intracellular pools in response to LPS and ATP? Although our current data do not distinguish between these possibilities, a recent study suggests that the α1 subunit of the Na⁺/K⁺-ATPase (NKAα1) forms a complex with P2X7 in microglia, including BV2 cells, and that LPS +ATP induces NKAα1 internalization (*Huang et al., 2024*). This internalization appears to release P2X7 from NKAα1, allowing P2X7 to exist in its free form. We speculate that the internalization of NKAα1 induced by both LPS and ATP exposes previously masked P2X7 sites, including the allosteric AZ10606120 sites, thus making them accessible for **X7-uP** labeling.

In sequence, the first priming step induced by LPS triggers P2X7 clustering and tends to increase the number of cell-surface receptors. This modest increase in P2X7 density may explain the lack of functional upregulation in the amplitude of the P2X7 current component observed in BV2 cells following LPS treatment (*Raouf et al., 2007*; *Trang et al., 2020*). The second step, triggered by ATP exposure, further elevates surface receptor levels and activates the P2X7-mediated NLRP3 inflammasome, which cleaves pro-IL-1β into its mature form and facilitates its release. Previous evidence suggests that P2X7 and NLRP3 co-localize in mouse microglia (*Franceschini et al., 2015*), indicating that NLRP3 may reside beneath P2X7 clusters. We speculate that the spatial reorganization and clustering of P2X7 during the LPS-induced priming phase occurs at sites of active inflammation, facilitating synergistic IL-1β secretion in response to ATP. Although additional experiments are needed to precisely colocalize P2X7 clusters with sites of active inflammation, this clustering mechanism might represent a finely tuned regulatory process that links PAMP-induced signaling to DAMP-evoked responses.

What mechanisms underlie P2X7 clustering in response to inflammatory signals? Several models have been proposed to explain membrane protein clustering, including recruitment to structural scaffolds (*Feng and Zhang, 2009*), partitioning into membrane domains enriched in specific chemical components such as lipid rafts (*Simons and Ikonen, 1997*), and self-assembly mechanisms (*Sieber et al., 2007*). These self-assembly mechanisms include an irreversible stochastic model (*Sato et al., 2019*) and a more recent reversible self-oligomerization model which gives rise to higher-order transient structures (HOTS; *Zhang et al., 2025*). Supported by cryogenic optical localization microscopy with very high resolution (~5 nm), the HOTS model has been observed in various membrane proteins, including ion channels and receptors (*Zhang et al., 2025*). Furthermore, HOTS are suggested to be dynamically modulated and to play a functional role in cell signaling, potentially influencing both physiological and pathological processes (*Zhang and MacKinnon, 2025*). While this hypothesis is compelling, our current dSTORM data lack sufficient spatial resolution to confirm whether P2X7 trimers form HOTS via self-oligomerization. Further biophysical and ultra-high-resolution imaging studies are required to test this model in the context of P2X7 clustering.

In conclusion, **X7-uP** is a powerful tool for visualizing P2X7 localization at the nanoscale on the cell surface of BV2 cells. We anticipate that **X7-uP** will become a valuable molecular probe for investigating P2X7 organization in other native cell types, such as macrophages, and for resolving the ongoing debate about P2X7 expression in neurons (*Illes et al., 2017*; *Miras-Portugal et al., 2017*).

## Materials and methods

### Chemical synthesis

All chemicals were purchased from Sigma-Aldrich, Acros Organics, or Alfa Aesar in analytical grade. An Agilent MM-ESI-ACI-SQ MSD 1200 SL spectrometer or an Agilent LC-MS Agilent RRLC 1200SL/ESI QTof 6520 was used for ESI analysis. $^{1}$H NMR and $^{13}$C NMR spectra were recorded at 400 or 700, and 100 or 175 MHz, respectively, using the following NMR Bruker instruments: a 400 MHz spectrometer equipped with an Avance III console and a BBO H/F z-gradient probe or a 700 MHz spectrometer equipped with a TCI z-gradient cryoprobe and an Avance III-HD console. Coupling constants (*J*) are quoted in hertz (Hz) and chemical shifts (δ) are given in parts per million (ppm) using the residue solvent peaks (CDCl$_3$: 7.26 ppm, MeOD: 3.31 ppm, DMSO-d$_6$: 2.50 ppm for $^{1}$H NMR and CDCl$_3$: 77.16 ppm, MeOD: 49 ppm, DMSO-d$_6$: 39.52 ppm for $^{13}$C NMR). The attributions are given in the following manner: chemical shift followed by the multiplicity in parenthesis (s, d, t, q, m, dd, dt, br corresponding respectively to singlet, doublet, triplet, quadruplet, multiplet, doublet of a doublet, doublet of a triplet, broad; number of protons and coupling constant in Hz).

HPLC analyses were performed on a Waters high-performance chromatography system (1525 pump, 2996 detector) equipped with a Thermo Betabasics 5 micron analytical column (4.6, 250 nm). A gradient solution was applied, progressing from 100% mQ H$_2$O acidified with 0.01% TFA to 100% acetonitrile over 30 min, followed by 10 min at 100% acetonitrile. HPLC purifications were carried out on a Waters high-performance chromatography system (600 double body pump, 2996 detector) equipped with a Thermo Betabasics 5 micron semi-preparative column (10, 250 nm), using the same gradient as described above.

**Chemical structure 1.** 5-aminoquinolin-2(1H)-one (3).

To a solution of 5-nitroquinolin-2(1H)-one (**2**) (1 g, 5.26 mmol, 1 eq.) dissolved in N,N-dimethylformamide (DMF, 40 mL), frozen at -196°C, was added dry palladium on activated charcoal (10%) (448 mg, 2.21 mmol, 0.8 eq.). The solution was allowed to warm to room temperature, and dihydrogen was bubbled in the solution for 5 minutes. The reaction was then maintained under dihydrogen atmosphere and stirred for 16 hours. The crude product was then filtered through a celite pad. The pad was further rinsed with DMF (100 mL) and the filtration product was evaporated to yield a grey solid (825 mg, 98 %).

$^1$H NMR (400 MHz, MeOD): δ 8.19 (dd, $J$=9.7, 1.0 Hz, 1 H), 7.26 (m, 1 H), 6.63 (dt, $J$=8.1, 1.0 Hz, 1 H), 6.55 (dd, $J$=8.1, 1.0 Hz, 1 H), 6.47 (d, $J$=9.7 Hz, 1 H). See *Figure 1—figure supplement 3* for spectrum.

**Chemical structure 2.** 2-(adamantan-1-yl)acetyl chloride (4).

2-(1-adamantyl)acetic acid (1.1 g, 5.67 mmol, 1 eq.) was placed in an oven-dried flask equipped with a reflux. Thionyl chloride (10 mL) was added, and the solution was heated to reflux under argon atmosphere for 3 hr. The solvent was then evaporated, and the resulting yellow oil was further dried under vacuum. The obtained compound was used immediately in the next step.

**Chemical structure 3.** 2-(adamantan-1-yl)-N-(2-oxo-1,2-dihydroquinolin-5-yl)acetamide (5).

To a stirred solution of 5-aminoquinolin-2(1 H)-one (**3**) (825 mg, 5.15 mmol, 1 eq.) in anhydrous tetrahydrofuran (THF) was added freshly distilled triethylamine (1.11 mL, 8.24 mmol, 1.6 eq.). The mixture was cooled to 0 °C and a solution of 2-(-adamantan-1-yl)acetyl chloride (1.2 g, 5.67 mmol, 1.1 eq) (**4**) dissolved in dry THF (5 mL) was added dropwise over the course of 15 min to form a gray solution. The solution was kept at 0 °C for 1 hr and brought to 20 °C for 2 more hours. The solution was then evaporated and the obtained solid was rinsed three times with CH$_2$Cl$_2$ (50 mL). The solid was further dried to yield the expected compound as a gray solid (1.4 g, 81 %). The product was then used without further purification.

$^1$H NMR (400 MHz, MeOD): δ 8.06 (dd, $J$=9.8, 0.8 Hz, 1 H), 7.58–7.54 (m, 1 H), 7.33 (dd, $J$=7.8, 1.1 Hz, 1 H), 7.26 (m, 1 H), 6.65 (d, $J$=9.8 Hz, 1 H), 2.26 (s, 2 H), 2.03 (m, 3 H), 1.86–1.70 (m, 12 H). See *Figure 1—figure supplement 4* for spectrum.

**Chemical structure 4.** 2-(adamantan-1-yl)-N-(2-chloroquinolin-5-yl)acetamide (6).

To a stirred solution of 2-(adamantan-1-yl)-*N*-(2-oxo-1,2-dihydroquinolin-5-yl)acetamide (**5**) (725 mg, 2.15 mmol, 1 eq.) dissolved in 1,2-dichloroethane (10 mL) was added freshly distilled POCl₃ (618 µL, 6.45 mmol, 3 eq.). The solution was brought to reflux for 16 hr, and the solvent was evaporated. The crude product was dissolved in CH₂Cl₂, filtrated, and evaporated to yield the expected compound as an orange solid (760 mg, 99 %).

¹H NMR (400 MHz, MeOD): δ 8.43 (dd, *J*=8.8, 0.8 Hz, 1 H), 7.87–7.79 (m, 2 H), 7.72 (dd, *J*=7.2, 1.5 Hz, 1 H), 7.59–7.55 (m, 1 H), 2.30 (s, 2 H), 2.03 (m, 3 H), 1.86–1.72 (m, 12 H). See *Figure 1—figure supplement 5* for spectrum.

**Chemical structure 5.** 2-(adamantan-1-yl)-N-(2-((2-aminoethyl)amino)quinolin-5-yl)acetamide (7).

To a stirred solution of 2-(adamantan-1-yl)-N-(2-chloroquinolin-5-yl)acetamide (**6**) (488 mg, 1.36 mmol, 1 eq.) in freshly distilled ethanol (10 mL) was added dry K₂CO₃ (376 mg, 2.72 mmol, 2 eq.) and anhydrous ethylene diamine (4.5 mL, 68 mmol, 50 eq.). The mixture was stirred at reflux for 48 hours. The solution was evaporated and subsequently extracted with 3 × 50 mL CH₂Cl₂. Organic layers were combined, washed with 25 mL of brine, and dried over MgSO₄, filtered, and evaporated. The crude extract was dissolved in a minimum of CH₂Cl₂ and diethyl ether was added dropwise until the apparition of opalescence. Heptane was then slowly added to precipitate the pure product (262 mg, 51 %) as a yellow solid.

¹H NMR (400 MHz, MeOD): δ 7.96 (d, *J*=9.2 Hz, 1 H), 7.56–7.46 (m, 2 H), 7.23 (dd, *J*=7.2, 1.4 Hz, 1 H), 6.80 (d, *J*=9.2 Hz, 1 H), 3.57 (t, *J*=6.2 Hz, 2 H), 2.91 (t, *J*=6.2 Hz, 2 H), 2.25 (s, 2 H), 2.03 (m, 3 H), 1.87–1.68 (m, 12 H). See *Figure 1—figure supplement 6* for spectrum.

**Chemical structure 6.** N-(2-((5-(2-(adamantan-1-yl)acetamido)quinolin-2-yl)amino)ethyl)–4-sulfamoylbenzamide (8).

To a stirred solution of 2-(adamantan-1-yl)-N-(2-((2-aminoethyl)amino)quinolin-5-yl)acetamide (**7**) (105 mg, 0.28 mmol, 1 eq.) in dry DMF (2.5 mL) were added 4-sulfamoylbenzoic acid (67 mg, 0.33 mmol, 1.2 eq.), 1-Hydroxybenzotriazole (45 mg, 0.33 mmol, 1.2 eq.) and DIEA (129 µL, 0.75 mmol, 2.7 eq.). The solution was cooled to 0 °C and EDC (64 mg, 0.33 mmol, 1.2 eq.) was added. The mixture

was then slowly warmed to 20 °C and stirred at 20 °C for 18 hr. The solvent was evaporated and the crude product was purified by invert phase flash chromatography (gradient acetonitrile: H$_2$O (0.1% TFA), 0:1 to 1:0 over 30 min, retention time: 16.2 min) to yield pure compounds as an orange solid (100 mg, 64 %).

$^1$H NMR (400 MHz, MeOD): δ 8.19 (d, J=9.5 Hz, 1 H), 8.02–7.91 (m, 4 H), 7.70 (m, 2 H), 7.45 (dd, J=7.5, 1.3 Hz, 1 H), 7.04 (d, J=9.5 Hz, 1 H), 3.83 (t, J=6.2 Hz, 2 H), 3.73 (t, J=6.2 Hz, 2 H), 2.28 (s, 2 H), 2.02 (s, 3 H), 1.88–1.68 (m, 12 H).

$^{13}$C NMR (100 MHz, MeOD): δ 173.92, 169.94, 165.28, 160.81, 156.37, 148.36, 138.62, 136.24, 133.03, 129.52, 127.77, 122.82, 119.63, 119.19, 114.59, 65.67, 55.16, 52.36, 44.52, 38.31, 34.83, 30.50. See *Figure 1—figure supplement 7* for spectra.

MS (ESI): m/z [M$^+$] calculated for C$_{30}$H$_{36}$N$_5$O$_4$S$^+$: 562.2483, found 562.2516.

**Chemical structure 7.** tert-butyl (15-((4-(2-((5-(2-(adamantan-1-yl)acetamido)quinolin-2-yl)amino)ethyl)carbamoyl) phenyl)sulfonamido)–15-oxo-3,6,9,12-tetraoxapentadecyl)carbamate (10).

To a stirred solution of *N*-(2-((5-(2-(adamantan-1-yl)acetamido)quinolin-2-yl)amino)ethyl)–4-sulfamoylbenzamide (**8**) (43 mg, 76 μmol, 1 eq.) in dry DMF (1 mL) were added COOH-OEG4-Boc (**9**) (33.6 mg, 92 μmol, 1.2 eq.), DIEA (33 μL, 191 μmol, 2.5 eq.) and DMAP (2 mg, 15 μmol, 0.2 eq.). The solution was cooled to 0 °C, and EDC (17.6 mg, 92 μmol, 1.2 eq.) was added. The reaction mixture was then slowly warmed to 20 °C and stirred at 20 °C for 16 hr. Every 24 hr over a total of 72 hr, an additional solution of 1 eq. of carboxylic acid, 1 eq. of EDC, and 2 eq. of DIEA prepared at 0 °C in dry DMF (1 mL) was added. The solvent was then evaporated, and 50 mL of a 1:1 CH$_2$Cl$_2$:H$_2$O solution was added. The organic layer was extracted with 2 × 20 mL of CH$_2$Cl$_2$. Organic layers were then combined, dried over MgSO$_4$, filtered, and evaporated. The crude product was purified by HPLC purification (acetonitrile:H$_2$O (0.1% TFA), 0:1 to 1:0 over 30 minutes, retention time: 17.7 min). Pure fractions were combined, extracted with 3 × 20 mL of CH$_2$Cl$_2$, dried over MgSO$_4$, and evaporated to yield pure compound as a dark orange oil (57 mg, 82%).

$^1$H NMR (400 MHz, CDCl$_3$): δ 8.20 (m, 1 H), 8.01 (m, 4 H), 7.42 (m, 3 H), 6.85 (s, 1 H), 3.79 (m, 2 H), 3.60 (m, 18 H), 3.26 (s, 2 H), 2.48 (s, 2 H), 2.22 (s, 2 H), 1.98 (s, 3 H), 1.77–1.59 (m, 12 H), 1.41 (s, 9 H).

$^{13}$C NMR (100 MHz, CDCl3): δ 171.08, 170.38, 168.12, 166.33, 156.37, 153.55, 139.80, 137.58, 134.55, 133.23, 128.28, 127.87, 121.02, 115.21, 113.45, 108.43, 79.43, 70.45, 70.38, 70.22, 70.13, 66.13, 53.42, 51.57, 42.68, 36.69, 33.47, 28.51. See *Figure 1—figure supplement 8* for spectra.

**Chemical structure 8.** N-(2-((5-(2-(adamantan-1-yl)acetamido)quinolin-2-yl)amino)ethyl)–4-(N-(17-oxo-21-((3aR,4R,6aS)–2-oxohexahydro-1H-thieno[3,4-d]imidazol-4-yl)–4,7,10,13-tetraoxa-16-azahenicosanoyl)sulfamoyl) benzamide (11).

To a stirred solution tert-butyl (15-((4-((2-((5-(2-(adamantan-1-yl)acetamido)quinolin-2-yl)amino)ethyl)carbamoyl)phenyl)sulfonamido)–15-oxo-3,6,9,12-tetraoxapentadecyl)carbamate (**10**) (7.6 mg, 8 μmol, 1 eq.) in 700 μl of CH₂Cl₂ was added 100 μl of TFA and the solution was left under Argon for 1 hr at 20 °C. After completing deprotection of the amine (confirmed by HPLC), the solvent was evaporated and the residue was further dried by three successive co-evaporation with anhydrous toluene (5 mL). Anhydrous DMF was then added to the flask (1 mL), followed by N-methylmorpholine (2 μL, 17 μmol, 2 eq.), biotin (3 mg, 13 μmol, 1.5 eq.), and 4-(4,6-dimethoxy-1,3,5-triazin-2-yl)–4-methyl-morpholinium chloride (DMTMM, 3.4 mg, 13 μmol, 1.5 eq.). The solution was left 2 hr at 20 °C. The solvent was then evaporated and the pure product was obtained by HPLC purification (acetonitrile: H₂O (0.1% TFA), 0:1 to 1:0 over 30 min, retention time: 18.1 min) to yield an orange solid (7 mg, 80%).

¹H NMR (500 MHz, MeOD): δ 10.01 (s, 1 H), 9.01 (br, 1 H), 8.29 (s, 1 H), 8.07–7.96 (m, 4 H), 7.82 (t, J=5.6 Hz, 1 H), 7.76–7.69 (br, 1 H), 7.60 (br, 1 H), 7.09 (s, 1 H), 6.41 (s, 1 H), 6.36 (s, 1 H), 4.31 (dd, J=7.6, 4.2 Hz, 1 H), 4.13 (dd, J=7.6, 4.4 Hz, 1 H), 3.76 (br, 2 H), 3.64 (br, 2 H), 3.55–3.35 (m, 16 H), 3.18 (q, J=6.2 Hz, 2 H), 3.12–3.07 (m, 1 H), 2.82 (dd, J=12.6, 5.3 Hz, 1 H), 2.58 (J=12.6, 2 H), 2.21 (m, 2 H), 2.06 (t, J=7.3 Hz, 2 H), 1.97 (s, 3 H) 1.72–1.57 (m, 12 H), 1.53–1.41 (m, 3 H), 1.35–1.23 (m, 3 H).

¹³C NMR (100 MHz, MeOD): δ 174.74, 172.09, 170.58, 168.23, 164.72, 159.59, 153.15, 137.88, 136.88, 135.05, 132.40, 128.07, 127.70, 122.07, 116.79, 115.21, 113.51, 70.14, 70.12, 70.00, 69.17, 65.80, 61.98, 60.26, 55.58, 50.55, 42.46, 39.67, 38.97, 36.48, 35.33, 33.05, 28.79, 28.33, 28.09, 25.44. See *Figure 1—figure supplement 9* for spectra.

**Chemical structure 9.** N-(2-((5-(2-(adamantan-1-yl)acetamido)quinolin-2- yl)amino)ethyl)–4-(N-(cyanomethyl)-N-(17-oxo-21-((3aR,4R,6aS)–2-oxo- hexahydro-1H-thieno[3,4-d]imidazol-4-yl)–4,7,10,13-tetraoxa-16-azahenico- sanoyl)sulfamoyl)benzamide (X7-uP).

To a stirred solution of N-(2-((5-(2-(adamantan-1-yl)acetamido)quinolin-2-yl)amino)ethyl)–4-(N-(17-oxo-21-((3aR,4R,6aS)–2-oxohexahydro-1H-thieno[3,4-d]imidazol-4-yl)–4,7,10,13-tetraoxa-16-azahenicosanoyl)sulfamoyl)benzamide (**11**) (4 mg, 4 μmol, 1 eq.), previously dried with five successive co-evaporation with anhydrous toluene (5 mL), in dried DMF (0.5 mL) were added freshly distilled DIEA (32 μl, 190 μmol, 50 eq.) and iodoacetonitrile (13 μl, 190 μmol, 50 eq.). The iodoacetonitrile was dried over an alumina plug prior to use. The solution was stirred for 16 hr at 20 °C in the dark, and the solvent was then evaporated. The crude product was purified by HPLC (acetonitrile: H₂O (0.1% TFA), 0:1 to 1:0 over 30 min, retention time: 19.1 min), affording the pure compound as a brown solid (3.3 mg, 80 %).

¹H NMR (700 MHz, DMSO): δ 10.02 (s, 1 H), 9.08 (br, 1 H), 8.29 (s, 1 H), 8.14 (d, *J*=8.3 Hz, 2 H), 8.08 (d, *J*=8.3 Hz, 2 H), 7.83 (t, *J*=5.6 Hz, 1 H), 7.76–7.72 (br, 1 H), 7.60 (br, 1 H), 7.09 (s, 1 H), 6.42 (s, 1 H), 6.37 (s, 1 H), 5.02 (s, 2 H), 4.31 (dd, *J*=7.8, 5.0 Hz, 1 H), 4.13 (dd, *J*=7.7, 4.4 Hz, 1 H), 3.77 (s, 2 H), 3.66 (s, 2 H), 3.60 (t, *J*=6.1 Hz, 2 H), 3.53–3.42 (m, 12 H), 3.39 (t, *J*=6.0 Hz, 2 H), 3.18 (q, *J*=6.0 Hz, 2 H), 3.12–3.07 (m, 1 H), 2.94 (t, *J*=6.1 Hz, 2 H), 2.82 (dd, *J*=12.6, 5.1 Hz, 1 H), 2.58 (d, *J*=12.6 Hz, 2 H), 2.22 (m, 2 H), 2.07 (t, *J*=7.4 Hz, 2 H), 1.97 (s, 3 H), 1.73–1.58 (m, 12 H), 1.53–1.43 (m, 4 H), 1.34–1.23 (m, 2 H).

¹³C NMR (175 MHz, DMSO): δ 170.91, 169.04, 168.67, 164.13, 161.47, 161.06, 156.95, 156.77, 138.73, 137.84, 133.90, 129.02, 128.80, 127.17, 126.67, 116.81, 116.40, 114.96, 114.70, 68.52, 68.47, 68.44, 68.42, 68.31, 68.27, 63.96, 59.78, 57.94, 54.15, 53.63, 48.88, 40.83, 38.72, 37.15, 35.13, 34.63, 34.52, 33.82, 32.69, 31.52, 29.50, 29.42, 26.92, 26.78, 26.76, 23.99. See *Figure 1—figure supplement 10* for spectra.

MS (ESI): m/z [M+] calculated for C₅₃H₇₁N₉O₁₁S₂⁺: 1074.4787, found 1074.4797.

## Molecular modeling

The P2X7 structure resolved with the allosteric inhibitor AZ10606120 was retrieved from the Protein Data Bank (PDB, code 5U1W, *Karasawa and Kawate, 2016*). The inhibitor was removed, and docking simulations were performed using AutoDock Vina 1.5.6. A 30 Å box centered on the quinoline core of AZ10606120 was used for docking, set before ligand removal. The cartoon shown in *Figure 6* was created using tetrameric biotin-bound streptavidin (PDB code 1MK5, *Hyre et al., 2006*) and rP2X7 (PDB code 6U9W, *McCarthy et al., 2019*). Tetrameric biotin-bound streptavidin was positioned relative to K82, using the maximum 18.5 Å distance between the α-carbon of K82 and the carboxylate carbon of the bound biotin.

## Cell culture and transfection

HEK293T cells (ATCC CRL-3216; authenticated by STR profiling) and mouse BV2 microglia cells (RRID:CVCL_0182 from Banca-Biologica e Cell Factory N° ATL03001, a kind gift from Dr. R. Schlichter, Institute of Cellular and Integrative Neuroscience, University of Strasbourg, France; authenticated by STR profiling) were cultured in high-glucose Dulbecco's modified Eagle's medium (DMEM 31966–021) supplemented with 10% (v/v) heat-inactivated fetal bovine serum (FBS), 100 units/mL penicillin, and 100 µg/mL streptomycin (Gibco Life Technologies, USA). Cells were grown at 37 °C in a humidified cell incubator with 5% $CO_2$. Both cell lines tested negative for mycoplasma contamination.

For transfection, cells were grown to 70–80% confluence, and the calcium phosphate precipitation method was employed. The cDNA encoding rat P2X7-mScarlet (*Bindels et al., 2017*), P2X1-GFP, P2X2-GFP, P2X3-GFP, P2X4-GFP, P2X5-GFP, or P2X6-GFP (all kind gifts from Dr. F. Rassendren, Institut de Génomique Fonctionelle, University of Montpellier, France) was contained within pcDNA3.1(+) plasmids (Invitrogen, USA). rP2X7c-myc was obtained as previously described (*Dunning et al., 2021*). 24 hr after transfection, the medium was replaced with fresh medium. For whole-cell patch-clamp experiments, cells were co-transfected with the rP2X7 construct (0.8 µg) and an eGFP (0.3 µg) which allowed to identify cells that had undergone efficient transfection. For biochemical experiments, each 100 mm dish was transfected with P2X7c-myc construct (10 µg). For confocal imaging and dSTORM experiments, HEK293T cells were transfected with 1 µg of the indicated construct.

## Mutagenesis

Site-directed mutations were introduced into the P2X7-mScarlet cDNA in the pcDNA3.1(+) using KAPA HiFi HotStart PCR kit (Cliniscience, France) as previously described (*Habermacher et al., 2016*). All mutations were confirmed by DNA sequencing.

## Electrophysiology patch-clamp

Electrophysiological recordings in whole-cell configuration were carried out as previously described (*Dunning et al., 2021*). Briefly, patch pipettes were pulled from borosilicate glass capillaries and microforged to yield a resistance of 3–5 MΩ. Cells were voltage-clamped to –60 mV using EPC10 USB amplifier (HEKA, Reutlingen, Germany), and data were recorded with PATCHMASTER software (version V2X90.5). Ligands were applied via a perfusion system, using three capillary tubes placed directly over the cell of interest.

2'(3')-O-(4-benzoylbenzoyl) adenosine 5'-triphosphate (BzATP) (triethylammonium salt, Alomone labs, Israel) was used as P2X7 agonist at a concentration of 10 µM. AZ10606120 and **X7-uP** were used at the indicated concentrations. BzATP was dissolved in Normal Extracellular Solution (NES), containing 140 mM NaCl, 2.8 mM KCl, 1 mM $CaCl_2$, 0.1 mM $MgCl_2$, 10 mM HEPES, and 10 mM glucose, with a pH of 7.32–7.33. AZ10606120 and **X7-uP** were prepared as concentrated stocks in DMSO and diluted in NES to the desired working concentration (<0.1% DMSO). Only one cell was patched per coverslip.

## Labeling reaction with X7-uP

**X7-uP** was stored as a DMSO stock solution (24.9 mM) and diluted in FBS-free DMEM to achieve the desired working concentration for labeling (<0.1% DMSO). After three washing steps with freshly prepared sterile-filtered PBS+ buffer (containing 137 mM NaCl, 2.68 mM KCl, 10 mM $Na_2HPO_4$, 1.76 mM $KH_2PO_4$, 1 mM $MgCl_2$ and 0.4 mM $CaCl_2$), cells were incubated with **X7-uP** diluted in FBS-free DMEM for the indicated time at 37 °C. Following incubation, cells were washed three times

with PBS+ buffer and treated as indicated. For confocal and dSTORM experiments, biotinylated cells were incubated with 1 μg/mL of Strept-A 647 (S21374, Invitrogen, USA) for 10 min in PBS+ buffer containing 1% bovine serum albumin (BSA) at 37 °C. After three additional washes with PBS+ buffer, cells were fixed as described below.

## Cell-surface protein biotinylation assay

Cell-surface biotinylation assay was performed 24 hr after transfection of HEK239T cells with different GFP-labeled P2X subtypes. Cells were washed three times in ice-cold PBS+ buffer (pH 8.0) and incubated in 2 mM sNHS-SS-Biotin in ice-cold PBS+ for 30 min under gentle agitation. The reaction was quenched with 20 mM Tris in PBS+ for 10 min. After three additional washes, cells were lysed for 90 min at 4 °C under thorough agitation in lysis buffer (20 mM HEPES, 100 mM NaCl, 5 mM EDTA, 1 % Triton-X-100, protease inhibitor tablets). Lysates were centrifuged at 10,000 rpm for 10 min at 4 °C, and the supernatant (input fraction) was collected. The cleared lysates were then incubated overnight at 4 °C with NeutrAvidin-agarose resin (Thermo Fisher) pre-washed with lysis buffer. The resin was subsequently washed three times with washing buffer (20 mM HEPES, 500 mM NaCl, 5 mM EDTA, 1% Triton-X-100, protease inhibitor tablets) and twice with lysis buffer. Bound proteins were resuspended in NuPAGE LDS sample buffer (Thermo Fisher) containing 70 mM DTT, boiled for 10 min, and separated on a NuPAGE Novex 4–12% Bis-Tris gel (Thermo Fisher, USA) in MOPS buffer. Transfer and western blotting were carried out as described below.

## Pull down, SDS-PAGE, and western blot

Cells were solubilized by vortexing in a lysis buffer (20 mM HEPES, 100 mM NaCl, 1% Triton-X-100, 5 mM EDTA, 1% protease inhibitor (Thermo Fisher, Waltham, MA, USA)) at 4 °C for 90 min. Following centrifugation (10,000 rpm, 10 min), supernatants were collected, mixed with the NuPAGE LDS loading buffer (Thermo Fisher) containing 70 mM dithiothreitol (DTT), boiled for 10 min, and run on a NuPAGE Novex Bis-Tris 4–12% (Thermo Fisher, USA) in MOPS buffer. Proteins were transferred to a nitrocellulose membrane using semi-wet transfer (TransBlot Turbo system, Bio-Rad, CA, USA), and the membrane was blocked for 30 min in TPBS buffer (PBS buffer supplemented with 1% milk powder, 0.5% BSA, and 0.05% Tween-20). The membrane was then incubated overnight at 4 °C under gentle agitation with primary antibodies: 1:2000 anti-GFP antibody (MA5-15256, Thermo Fisher), 1:500 anti-c-Myc mouse antibody (13–2500, Invitrogen, Thermo Fisher), 1:500 rabbit anti-P2X7 antibody (APR-008, Alomone Labs, Israel), or mouse 1:5000 anti-β-actin (A5441, Sigma Aldrich). Proteins were visualized using 1:10,000 horseradish peroxidase-conjugated secondary antibodies directed against mouse or rabbit (NA9310, GE Life Sciences and 31460, Invitrogen, respectively) and a chemiluminescent substrate (Amersham ECL Select Western Blotting Detection, GE Life Sciences, MA, USA) on an Imager 600 (Amersham, IL, USA).

For pull-down experiments, solubilized cells (as described above) were incubated overnight at 4 °C under gentle agitation with Pierce NeutrAvidin agarose resin (29200, Thermo Fisher). Samples preparation and Western blotting was carried out as described above.

## Coverslip preparation

For confocal imaging and super-resolution experiments, coverslips (0117580, No. 1.5 H, Marenfield, Germany) were cleaned by incubation in freshly made piranha solution ($H_2SO_4$:$H_2O_2$ 50% 3:1) at 60 °C for 30 min. The cleaning solution was replaced with fresh piranha solution, and the coverslips were sonicated for an additional 30 min. After discarding the piranha solution, coverslips were washed 10 times with ultra-pure water and sonicated in ultrapure water (10 × 3 min each) to ensure complete removal of any cleaning residue. Coverslips were then dispatched in 12-well plates and exposed to UV light for 30 min. For HEK293T cells, the coverslips were treated with poly-L-lysine for 1 hr at 37 °C and subsequently washed with ultra-pure water before use. For BV2 cells, no treatment with poly-L-lysine was performed.

## BV2 microglia stimulation

BV2 microglia cells were seeded on treated coverslips (as described above) in a 12-well plate containing standard DMEM medium with 10% FBS. After 24 hr, cells were incubated in DMEM medium with 1% FBS for 24 hr, either left untreated or treated with *Escherichia coli* O111:B4 LPS (1 μg/mL, L2630,

Sigma Aldrich, USA). After washing once with freshly prepared sterile-filtered PBS+, cells were incubated in PBS+, either in the absence (vehicle) or supplemented with 1 mM adenosine 5'-triphosphate (ATP disodium salt hydrate, A7699, Sigma Aldrich) for 30 min, 300 µM BzATP for 30 min, or 15 mM MβCD (Sigma Aldrich, USA) for 15 min at 37 °C in a humidified cell incubator with 5% $CO_2$. After treatments, supernatants were collected for the IL-1β ELISA assay, and cells were labeled with 1 µM **X7-uP** as described above.

### ELISA assay

IL-1β levels were measured using the IL-1β Mouse Uncoated ELISA Kit (ref. 88–7013, Invitrogen, USA) following the manufacturer's protocol. Developed plates were read using a SAFAS Monaco plate reader with SP2000 version 7 software.

### Cell fixation

Cells were fixed by 4% paraformaldehyde (PFA) in PBS, pH 8.0 for 20 min at 20 °C. Fixed cells were then washed three times and incubated with fresh PBS buffer for 3 × 5 min, followed by incubation in 100 mM glycine (in PBS buffer) for 15 min at 20 °C. After three washes, cells were stored at 4 °C until use. All washing steps were performed with freshly made, sterile-filtered PBS buffer.

### Cytochemistry

For confocal imaging, coverslips containing fixed cells were treated with 4 µg/mL of Hoechst 33342 pentahydrate (bis-benzimide) (H21491, Molecular Probes, Life Technologies, USA) in PBS buffer for 10 min. The staining solution was discarded, and coverslips were washed three times with PBS buffer, rinsed with ultra-pure water, mounted on microscopy slides with the Prolong Gold Antifade (P36930, Invitrogen, USA), and allowed to dry overnight in the dark before use.

For confocal imaging of BV2 cells shown in *Figure 5—figure supplement 4*, coverslips were rinsed with ultra-pure water, mounted on microscopy slides using the Prolong Diamond Antifade with DAPI (P36962, Invitrogen, USA), and allowed to dry overnight in the dark before use.

For dSTORM experiments in HEK293T cells, coverslips were incubated at room temperature for 6 min in a PBS solution containing 1:5000 fluorescent microspheres (Tetraspeck, Invitrogen, USA) as fiducial markers for lateral drift correction during image reconstruction, rinsed with ultra-pure water, mounted on microscopy slides on a Vectashield H-1000 (Vector Laboratories, USA) containing a 50 mM TRIS-glycerol (obtained by diluting 5% v/v TRIS pH 8.0 in glycerol and filtered on 0.22 µm filter) in a 1:4 ratio, and sealed using dental cement (Picodent, Germany; *Olivier et al., 2013*).

For dSTORM experiments in BV2 cells, coverslips were first incubated at room temperature for 6 min in a PBS solution containing 1:5000 fluorescent microspheres (Tetraspeck, Invitrogen, USA). The coverslips were then mounted in a Ludin chamber (Life Imaging Services), where a buffer composed of an oxygen scavenger (glucose oxidase) and a reducing agent (2-mercaptoethylamine) was added (600 µL). The Ludin chamber was sealed with an 18 mm coverslip to prevent oxygen exchange.

### Confocal imaging

Confocal imaging was captured with Leica SPE (63 x oil immersion objective, N.A. 1.4). Excitation ($\lambda$ exc) and emission ($\lambda$ em) wavelengths were as follows: DAPI, Hoechst ($\lambda$ exc = 405 nm, $\lambda$ em = 430–480 nm), GFP ($\lambda$ exc = 488 nm, $\lambda$ em = 500–545 nm), mScarlet ($\lambda$ exc = 561 nm, $\lambda$ em = 570–610 nm) and Alexa Fluor-647 ($\lambda$ exc 635 nm, $\lambda$ em = 650–700 nm).

### dSTORM imaging and reconstruction

dSTORM acquisitions in HEK293T cells were conducted on a homemade system built on a Nikon Eclipse Ti microscope, equipped with a 100 x oil immersion objective (N.A. 1.49; *Glushonkov et al., 2018*). A 642 nm laser line was used to excite the Alexa Fluor-647 fluorophore, set to a power of 134 mW (resulting in an intensity of 2.7 kW/cm²). Samples were imaged with an EM-CCD camera (Hamamatsu, ImagEM) maintained at –60 °C. Z-stabilization was ensured by the Nikon Perfect Focus System integrated into the microscope. Acquisition was performed in TIRF mode controlled using Micro-Manager 1.4.23 (*Edelstein et al., 2014*). Single-molecule localization was achieved by analyzing a stack of 15,000 images with the ThunderSTORM ImageJ plugin (*Ovesný et al., 2014*). The following parameters were used: image filtering – Difference-of-Gaussians filter (sigma 1 = 1.2 px, sigma 2 =

1.9 px), approximate localization of molecules: centroid of connected components (peak intensity threshold std (Wave.F1), sub-pixel localization of molecules: PSF: Integrated Gaussian (fitting radius: 6 px, fitting method: Weighted Least squares, initial sigma: 1.6 px)). Results were filtered by sigma, localization precision, and intensities values: 110 nm< sigma <220 nm, precision <25 nm and intensity <2000.

dSTORM acquisitions in BV2 cells were performed on a commercial LEICA DMi8 (Leica, Germany) inverted microscope equipped with anti-vibrational table (TMC, USA) to minimize drift, along with a Leica HC PL APO 100 X oil immersion TIRF objective (NA 1.47). For sample excitation, the microscope was equipped with a fiber-coupled laser launch composed of the following wavelengths: 405, 488, 532, 561, and 642 nm (Roper Scientific, Evry, France). Samples were imaged with an EMCCD camera (Teledyne Photometrics). Z stabilization was guaranteed by the Leica auto focus system integrated into the microscope. The 642 nm laser was used at a constant power to excite the Alexa Fluor-647 fluorophore, and the 405 nm laser was adjusted throughout the acquisition to control the level of single molecules per frame. Image acquisition was performed in TIRF mode controlled by MetaMorph software (Molecular Devices), with a 30 ms frame duration and a stack of 20,000 frames per acquisition on a 512 x 512-pixel ROI (pixel size = 160 nm).

Super-resolved images were reconstructed using the PALMTracer plugin for MetaMorph (*Butler et al., 2022*). The localizations were first extracted using a consistent intensity threshold across the entire dataset. Subsequently, super-resolved images were generated from these localizations with a pixel size of 40 nm. The density of localization was then extracted on each image, based on ROIs determined around the plasma membrane.

## Cluster size measurement

Density-based clustering was performed using SR-Tesseler software (*Levet et al., 2015*) with the following input parameters: density factor 1 = 1; density factor 2 = 0.7, max length = 250 nm and min locs = 10; max locs = 200, max distance = 2 μm. The data obtained were further analyzed with GraphPad Prism (version 8.0.2).

## Kinetic analysis of X7-uP labeling

The **X7-uP** labeling reaction follows a kinetic model in which a reversible ligand binding reaction (here **X7-uP**) precedes an irreversible chemical reaction, as described previously (*Tamura et al., 2018*). In the presence of a large excess of **X7-uP** relative to P2X7 sites, the pseudo-first-order reaction rate constant ($k_{app}$) is given by *Equation 1*:

$$f(t) = \exp\left(-k_{app}t\right) \tag{1}$$

where *t* is the labeling reaction time at a given **X7-uP** concentration. The relationship between $k_{app}$ and **X7-uP** concentration is described by *Equation 2*:

$$k_{app} = \frac{k_L}{1 + K_d/\left[\text{X7-uP}\right]} \tag{2}$$

where $K_d$ is the dissociation constant of **X7-uP** and $k_L$ is the rate constant of the irreversible chemical reaction.

## Data analysis

All experiments were conducted with at least three independent experiments. Statistical significance was assumed when the *P*-value was <0.05. Graphs were generated using RStudio (version 1.4.1717) and ggplot2 package (version 3.3.5) or GraphPad Prism.

Western blot quantification was performed using the ImageJ gel analysis tool.

For confocal data, membrane fluorescence intensities were manually selected using the ROI function of ImageJ (version 2.3.0/1.53 f). Data were then analyzed using in-house written R scripts, employing the following packages: ggplot2, reshape2 (version 1.4.4), readxl (version 1.3.1), dplyr (version 1.0.7), Rmisc (version 1.5), and naniar (version.6.1). For confocal data shown in *Figure 5— figure supplement 4*, membrane intensities were automatically selected. A threshold was first applied to distinguish particles from the background, after which all particles >20 pixels in size were automatically selected and combined for each cell. After manually adjusting the selections, the integrated

density (mean gray value x pixel number) was determined using ImageJ's Measure function. Data were then analyzed with GraphPad.

For cluster analysis, data extracted from SR-Tesseler (cluster diameter, number of detections per cluster, inter-cluster distance) were computed with GraphPad Prism for each cell.

## Acknowledgements

We are grateful to Pr. P Didier for training on dSTORM in HEK293T cells, Dr. F Rassendren for providing P2X7-mScarlet and P2X-GFP constructs, Pr. R Schlichter for providing BV2 cells, and the IINS PBC facility and Dr. M Sainlos for providing mSA. This work was supported by the Agence Nationale de la Recherche (Grant ANR-20-CE14-0016-02), the Ministère de la Recherche (PhD grant to BA), the International Center for Frontier Research in Chemistry (Labex CSC-TGR-18) (PhD grant to FC), and The Région Grand Est (PhD grant to FC). This work also benefited from support provided by the University of Strasbourg Institute for Advanced Study (USIAS) for a fellowship, within the French national program "Investments for the Future" (Idex-Unistra) and from the Interdisciplinary Thematic Institute NeuroStra, as part of the ITI 2021–2028 program of the University of Strasbourg, CNRS and Inserm, supported by EUR (ANR-17-EURE-0022) (PhD grant extension to BA) under the framework of the French Program "Investments for the Future".

## Additional information

### Funding

| Funder | Grant reference number | Author |
|---|---|---|
| Agence Nationale de la Recherche | ANR-20-CE14-0016-2 | Thomas Grutter |
| International Center for Frontier Research in Chemistry | Labex CSC-TGR-18 | Francisco Andrés Peralta Thomas Grutter |
| International Center for Frontier Research in Chemistry (Labex CSC-TGR-18) and Région Grand Est | PhD grant | Federico Cevoli Thomas Grutter |
| Ministere de la Recherche | PhD grant | Benoit Arnould |
| University of Strasbourg Institute for Advanced Study | USIAS 2019 | Francisco Andrés Peralta Thomas Grutter |
| Interdisciplinary Thematic Institute NeuroStra | ANR-17-EURE-0022 | Benoit Arnould Thomas Grutter |

The funders had no role in study design, data collection and interpretation, or the decision to submit the work for publication.

### Author contributions

Benoit Arnould, Formal analysis, Investigation, Methodology, Writing – review and editing; Adeline Martz, Formal analysis, Investigation, Methodology; Pauline Belzanne, Data curation, Formal analysis, Investigation, Methodology; Francisco Andrés Peralta, Formal analysis, Investigation; Federico Cevoli, Investigation; Volodya Hovhannisyan, Yannick Goumon, Resources; Eric Hosy, Data curation, Formal analysis, Supervision, Writing – review and editing; Alexandre Specht, Formal analysis, Supervision, Writing – review and editing; Thomas Grutter, Conceptualization, Formal analysis, Supervision, Funding acquisition, Visualization, Methodology, Writing – original draft, Project administration, Writing – review and editing

### Author ORCIDs

Adeline Martz ⬤ https://orcid.org/0009-0008-4751-2587
Francisco Andrés Peralta ⬤ https://orcid.org/0000-0002-0727-1706

Eric Hosy ⬥ https://orcid.org/0000-0002-2479-5915
Thomas Grutter ⬥ https://orcid.org/0000-0002-4351-9203

Reviewer #1 (Public review): https://doi.org/10.7554/eLife.106096.3.sa1
Reviewer #2 (Public review): https://doi.org/10.7554/eLife.106096.3.sa2
Author response https://doi.org/10.7554/eLife.106096.3.sa3

## Additional files

### Supplementary files
MDAR checklist

### Data availability
All data generated or analyzed during this study are included in the manuscript, figures, figure supplements and source data files. RStudio codes to generate histograms of *Figure 3C*, *Figure 4C* and *Figure 5—figure supplement 1D* are provided in the source data files.

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
