## [Editor Report · eLife Assessment]

The authors employ an unbiased, affinity-guided reagent to label P2X7 receptor and use super-resolution imaging to monitor P2X7 redistribution in response to inflammatory signaling. The evidence is **convincing** and the study will be **valuable** to those studying the dynamics of receptor distribution and clustering.

---

## [Referee Report · Reviewer #1 (Public review)]

Summary:

In this paper, the authors developed a chemical labeling reagent for P2X7 receptors, called X7-uP. This labeling reagent selectively labels endogenous P2X7 receptors with biotin based on ligand-directed NASA chemistry. After labeling the endogenous P2X7 receptor with biotin, the receptor can be fluorescently labeled with streptavidin-AlexaFluor647. The authors carefully examined the binding properties and labeling selectivity of X7-uP to P2X7, characterized the labeling site of P2X7 receptors, and demonstrated fluorescence imaging of P2X7 receptors. The data obtained by SDS-PAGE, Western blot, and fluorescence microscopy clearly shows that X7-uP labels the P2X7 receptor. Finally, the authors fluorescently labeled the endogenous P2X7 in BV2 cells, which are a murine microglia model, and used dSTORM to reveal a nanoscale P2X7 redistribution mechanism under inflammatory conditions at high resolution.

Strengths:

X7-uP selectively labels endogenous P2X7 receptors with biotin. Streptavidin-AlexaFluor647 binds to the biotin labeled to the P2X7 receptor, allowing visualization of endogenous P2X7 receptors.

---

## [Referee Report · Reviewer #2 (Public review)]

Summary:

In this manuscript, Arnould et. al. develop an unbiased, affinity-guided reagent to label P2X7 receptor and use super-resolution imaging to monitor P2X7 redistribution in response to inflammatory signaling.

Strengths:

I think the X7-uP probe that they developed is very useful for visualizing localization of P2X7 receptor. They convincingly show that under inflammatory conditions, there is a reorganization of P2X7 localization into receptor clusters. Moreover, I think they have shown a very clever way to specifically label any receptor of interest. This has broad appeal.

I think the authors have done a very nice job addressing my original concerns. Here are those original concerns and my new comments related to how the authors address them.

(1) While the authors state that chemical modification of AZ10606120 to produce the X7-UP reagent has "minimal impact" on the inhibition of P2X7, we can see from Figure 2A and 2B that it does not antagonize P2X7 as effectively as the original antagonist. For the sake of completeness and quantitation, I think it would be great if the authors could determine the IC50 for X7-uP and compare it to the IC50 of AZ10606120.

The authors now show the relative inhibition of X7-uP compared to AZ10606120 at different concentrations. This provides a nice comparison to give the reader an idea of how effectively X7-uP inhibits P2X7 receptor. This is great.

(2) Do the authors know whether modification of the lysines with biotin affects the receptor's affinity for ATP (or ability to be activated by ATP)? What about P2X7 that has been modified with biotin and then labeled with Alexa 647? For the sake of completeness and quantitation, I think it would be great if the authors could determine the EC50 of biotinylated P2X7 for ATP as well as biotinylated and then Alexa 647 labeled P2X7 for ATP and compare these values to the affinity of unmodified WT P2X7 for ATP.

I agree with the authors that assessing the functional integrity of P2X7 following biotinylation and fluorophore labeling is outside the scope of this paper but would be important for studies involving dynamic or post-labeling functional analyses such as live trafficking.

(3) It is a little misleading to color the fluorescence signal from mScarlet green (for example, in Figure 3 and Figure 4). The fluorescence is not at the same wavelength as GFP. In fact, the wavelength (570 nm - 610 nm) for emission is closer to orange/red than to green. I think this color should be changed to differentiate the signal of mScarlet from the GFP signal used for each of the other P2X receptor subtypes.

The authors have now changed the mScarlet color to orange, which solves my concern.

(4) It is my understanding that P2X6 does not form homotrimers. Thus, I was a little surprised to see that the density and distribution of P2X6-GFP in Figure 3 looks very similar to the density and distribution of the other P2X subtypes. Do the authors have an explanation for this? Are they looking at P2X6 protomers inserted into the plasma membrane? Does the cell line have endogenous P2X receptor subtypes? Is Figure 3 showing heterotrimers with P2X6 receptor? A little explanation might be helpful.

The authors address this point very well and include nice data to show that P2X6 does not insert into the plasma membrane as a homotrimer.

(5) It is easy to overlook the fact that the antagonist leaves the binding pocket once the biotin has been attached to the lysines. It might be helpful if the authors made this a little more apparent in Figure 1 or in the text describing the NASA chemistry reaction.

The authors have modified Figure 1 to make it easier to understand the NASA chemistry reaction.

I congratulate the authors on an outstanding paper!

---

## [Author Response]

The following is the authors’ response to the original reviews.

**Public Reviews:**

**Reviewer #1 (Public review):**
Summary:In this paper, the authors developed a chemical labeling reagent for P2X7 receptors, called X7-uP. This labeling reagent selectively labels endogenous P2X7 receptors with biotin based on ligand-directed NASA chemistry (Ref. 41). After labeling the endogenous P2X7 receptor with biotin, the receptor can be fluorescently labeled with streptavidin-AlexaFluor647. The authors carefully examined the binding properties and labeling selectivity of X7-uP to P2X7, characterized the labeling site of P2X7 receptors, and demonstrated fluorescence imaging of P2X7 receptors. The data obtained by SDS-PAGE, Western blot, and fluorescence microscopy clearly show that X7-uP labels the P2X7 receptor. Finally, the authors fluorescently labeled the endogenous P2X7 in BV2 cells, which are a murine microglia model, and used dSTORM to reveal a nanoscale P2X7 redistribution mechanism under inflammatory conditions at high resolution.Strengths:X7-uP selectively labels endogenous P2X7 receptors with biotin. Streptavidin-AlexaFluor647 binds to the biotin labeled to the P2X7 receptor, allowing visualization of endogenous P2X7 receptors.

We thank the reviewer for their positive comment.

Weaknesses:Weaknesses & Comments(1) The P2X7 receptor exists in a trimeric form. If it is not a monomer under the conditions of the pull-down assay in Figure 2C, the quantitative values may not be accurate.

We thank the reviewer for this comment. As shown in Figure 2C, the band observed on the denaturing SDS-PAGE corresponds to the monomeric form of the P2X7 receptor. While we cannot exclude the presence of non-monomeric species under native conditions, no such higher-order forms are visible in the gel. This observation supports the conclusion that the quantitative values presented are based on the monomeric form and are therefore reliable.

(2) In Figure 3, GFP fluorescence was observed in the cell. Are all types of P2X receptors really expressed on the cell surface ?

We thank the reviewer for this excellent comment, which was also raised by reviewer 2. To address this concern, we performed a commercial cell-surface protein biotinylation assay to assess whether GFP-tagged P2X receptors reach the plasma membrane. As expected, all P2X subtypes except P2X6 were detected at the cell surface in HEK293T cells, thereby validating our confocal fluorescence microscopy assay. These new data are now included in Figure 3 — figure supplement 1.

(3) The reviewer was not convinced of the advantages of the approach taken in this paper, because the endogenous receptor labeling in this study could also be done using conventional antibody-based labeling methods.

We thank the reviewer for raising this important point and would like to highlight several advantages of our approach compared to conventional antibody-based labeling.

First, commercially available P2X7 antibodies often suffer from poor specificity and are generally not suitable for reliably detecting endogenous P2X7 receptors, as documented in previous studies (e.g., PMID: 16564580 and PMID: 15254086). While recent advances have been made using nanobodies with improved specificity for P2X7 (e.g., PMID: 30074479 and PMID: 38953020), our strategy is distinct and complementary to nanobody-based approaches.

Second, antibodies rely on non-covalent interactions with the receptor, which can result in dissociation over time. In contrast, our X7-uP probe covalently biotinylates lysine residues on the P2X7 receptor through stable amide bond formation. This covalent labeling ensures that the biotin moiety remains permanently attached, an advantage not afforded by reversible binding strategies.

Third, by selectively biotinylating P2X7 receptors, our method provides a versatile platform for the chemical attachment of a wide range of probes or functional moieties. Although we did not demonstrate this application in the current study, we believe this modularity represents an additional advantage of our approach.

We have now revised the discussion to highlight these key advantages, allowing the reader to form their own opinion. We hope this addresses the reviewer’s concerns and clarifies the benefits of our approach.

(4) Although P2X7 was successfully labeled in this paper, it is not new as a chemistry. There is a need for more attractive functional evaluation such as live trafficking analysis of endogenous P2X7.

We agree with the reviewer that the underlying chemistry is not novel per se. However, to our knowledge, it has not previously been applied to the P2X7 receptor, and thus constitutes a novel application with specific relevance for studying native P2X7 biology.

We also appreciate the reviewer’s suggestion regarding live trafficking analysis of endogenous P2X7. While this is indeed a valuable and interesting direction, we believe it lies beyond the scope of the present study, as it would first require demonstrating that the labeling itself does not affect P2X7 function (see below). This important step would necessitate additional experiments, which we consider more appropriate for a follow-up investigation.

(5) The reviewer has concerns that the use of the large-size streptavidin to label the P2X7 receptor may perturbate the dynamics of the receptor.

We thank the reviewer for raising this important point. Although we did not directly measure receptor dynamics, it is indeed possible that tetrameric streptavidin (tStrept-A 647) could promote P2X7 clustering by cross-linking nearby receptors due to its tetravalency (see also point 7 raised by the reviewer). To address this concern, we performed additional dSTORM experiments using a monomeric form of streptavidin-Alexa 647 (mSA) (see PMID: 26979420). Owing to its reduced size and lack of tetravalency, mSA has been shown to minimize artificial crosslinking of synaptic receptors (PMID: 26979420). A drawback of using mSA, however, is that the monomeric form carries only two fluorophores (estimated degree of labeling, DOL ≈ 2, PMID: 26979420), whereas the tetrameric form, according to the manufacturer’s certificate of analysis (Invitrogen S21374), has an average DOL of three fluorophores per monomer, resulting in a total of ~12 fluorophores per streptavidin.

We tested three conditions with mSA incubation: (i) control BV2 cells (without X7-uP), (ii) untreated X7-uP-labeled BV2 cells, and (iii) X7-uP-labeled BV2 cells treated with LPS and ATP (using the same concentrations and incubation times described in the manuscript). As shown in Author response image 1, only LPS+ATP treatment induced a clear increase in the mean cluster density compared to quiescent (untreated) BV2 cells. This effect closely matches the results obtained with tStrept-A 647, supporting the conclusion the tetrameric streptavidin does not artificially promote P2X7 clustering. It is also possible that the cellular environment of BV2 microglia differs from the confined architecture of synapses, which may further explain why cross-linking effects are less pronounced in our system.

As expected, the overall fluorescence signal with mSA was about tenfold lower than with tStrept-A 647, consistent with the expected fluorophore stoichiometry. This lower signal may explain why the values for the untreated condition appeared slightly higher than for the control, although the difference was not statistically significant (P = 0.1455).

We hope these additional experiments adequately address the reviewer’s concerns.

**Author response image 1. sa3fig1:** BV2 labeling with monomeric streptavidin–Alexa 647 (mSA). (A) Bright-field and dSTORM images of BV2 cells labeled with mSA in the presence (untreated and LPS+ATP) or absence (control) of 1 µM X7-uP. Treatment: LPS (1 µg/mL for 24 hours) and ATP (1 mM for 30 minutes). Scale bars, 10 µm. Insets: Magnified dSTORM images. Scale bars, 1 µm.(B) Quantification of the number of localizations (n = 2 independent experiments). Bars represent mean ± s.e.m. One-way ANOVA with Tukey’s multiple comparisons (P values are indicated above the graph).

(6) It is better to directly label Alexa647 to the P2X7 receptor to avoid functional perturbation of P2X7.

Directly labeling of Alexa647 to the P2X7 receptor would require the design and synthesis of a novel probe, which is currently not available. Implementing such a strategy would involve substantial new experimental work that lies beyond the scope of the present study.

(7) In all imaging experiments, the addition of streptavidin, which acts as a cross-linking agent, may induce P2X7 receptor clustering. This concern would be dispelled if the receptors were labeled with a fluorescent dye instead of biotin and observed.

We refer the reviewer to our response in point 5, where we addressed this concern by comparing tetrameric and monomeric streptavidin conjugates. As noted above (see also point 6), directly labeling the receptor with a fluorescent dye would require the development of a new probe, which is outside the scope of the present study.

(8) There are several mentions of microglia in this paper, even though they are not used. This can lead to misunderstanding for the reader. The author conducted functional analysis of the P2X7 receptor in BV-2 cells, which are a model cell line but not microglia themselves. The text should be reviewed again and corrected to remove the misleading parts that could lead to misunderstanding. e.g. P8. lines 361-364First, it combines N-cyanomethyl NASA chemistry with the high-affinity AZ10606120 ligand, enabling rapid labeling in microglia (within 10 min)P8. lines 372-373Our results not only confirm P2X7 expression in microglia, as previously reported (6, 26-33), but also reveal its nanoscale localization at the cell surface using dSTORM.

We agree with the reviewer’s comment. We have now modified the text, including the title.

**Reviewer #2 (Public review):**
Summary:In this manuscript, Arnould et. al. develop an unbiased, affinity-guided reagent to label P2X7 receptor and use super-resolution imaging to monitor P2X7 redistribution in response to inflammatory signaling.Strengths:I think the X7-uP probe that they developed is very useful for visualizing localization of P2X7 receptor. They convincingly show that under inflammatory conditions, there is a reorganization of P2X7 localization into receptor clusters. Moreover, I think they have shown a very clever way to specifically label any receptor of interest. This has broad appeal

We thank the reviewer for their positive comment.

Weaknesses:Overall, the manuscript is novel and interesting. However, I do have some suggestions for improvement.(1) While the authors state that chemical modification of AZ10606120 to produce the X7-UP reagent has "minimal impact" on the inhibition of P2X7, we can see from Figure 2A and 2B that it does not antagonize P2X7 as effectively as the original antagonist. For the sake of completeness and quantitation, I think it would be great if the authors could determine the IC50 for X7-uP and compare it to the IC50 of AZ10606120.

We thank the reviewer for this insightful comment. Unfortunately, due to the limited availability of X7-uP, we were not able to establish a complete concentration–response curve to determine its IC_50_, which would require testing at concentrations >1 µM. Nevertheless, to estimate the effect of the modification, we assessed current inhibition at 300 µM X7-uP and compared it with the reported IC_50_ of AZ10606120 (10 nM). Under these conditions, both compounds produced a similar level of inhibition, indicating that while the chemical modification reduces potency relative to AZ10606120, X7-uP still functions as an effective probe for P2X7. We have now included these data in Figure 2 and revised the text accordingly.

(2) Do the authors know whether modification of the lysines with biotin affects the receptor's affinity for ATP (or ability to be activated by ATP)? What about P2X7 that has been modified with biotin and then labeled with Alexa 647? For the sake of completeness and quantitation, I think it would be great if the authors could determine the EC50 of biotinylated P2X7 for ATP as well as biotinylated and then Alexa 647 labeled P2X7 for ATP and compare these values to the affinity of unmodified WT P2X7 for ATP.

We thank the reviewer for raising this important point. At present, we have not determined whether modification of lysine residues with biotin, or subsequent labeling with Alexa647, affects the ATP sensitivity or functional properties of P2X7. However, we believe this does not impact the conclusions of the current study, as all functional assays were conducted prior to X7-uP labeling. The labeling is used here as a terminal "snapshot" to visualize the endogenous receptor without interfering with the functional characterization.

We fully agree that assessing the functional integrity of P2X7 following biotinylation and fluorophore labeling—such as by determining the EC_50_ for ATP—would be essential for studies involving dynamic or post-labeling functional analyses, such as live trafficking. However, as noted earlier in our response to Reviewer 1 (point 4), these experiments lie beyond the scope of the current study.

(3) It is a little misleading to color the fluorescence signal from mScarlet green (for example, in Figure 3 and Figure 4). The fluorescence is not at the same wavelength as GFP. In fact, the wavelength (570 nm - 610 nm) for emission is closer to orange/red than to green. I think this color should be changed to differentiate the signal of mScarlet from the GFP signal used for each of the other P2X receptor subtypes.

As suggested, we changed the mScarlet color to orange for all relevant figures.

(4) It is my understanding that P2X6 does not form homotrimers. Thus, I was a little surprised to see that the density and distribution of P2X6-GFP in Figure 3 looks very similar to the density and distribution of the other P2X subtypes. Do the authors have an explanation for this? Are they looking at P2X6 protomers inserted into the plasma membrane? Does the cell line have endogenous P2X receptor subtypes? Is Figure 3 showing heterotrimers with P2X6 receptor? A little explanation might be helpful.

We thank the reviewer for raising this important point. Indeed, it is well established that P2X6 does not form functional channels, which supports the conclusion that it does not form homotrimeric complexes. Although previous studies have shown that P2X6–GFP expression is generally lower, more diffuse, and not efficiently targeted to the cell surface compared with other P2X subtypes (see PMID: 12077178), the similar fluorescence distribution and density observed in our Figure 3 do not imply that P2X6 forms homotrimers.

We did not directly assess the presence of endogenous P2X6 in our HEK293T cells; however, according to the Human Protein Atlas, there is no detectable P2X6 RNA expression in HEK293 cells (nTPM = 0), indicating that endogenous P2X6 is not expressed in this cell line. To further investigate surface expression (see also point 2 of reviewer 1), we performed a commercial cell-surface protein biotinylation assay to assess whether GFP-tagged P2X6 reaches the plasma membrane. As expected, P2X6 was not detected at the cell surface in HEK293T cells, whereas GFP-tagged P2X1 to P2X5 were readily detected. These results further support the conclusion that P2X6 does not insert into the plasma membrane as a homotrimer, thereby validating our confocal fluorescence microscopy assay. These new data are now included in Figure 3 — figure supplement 1.

(5) It is easy to overlook the fact that the antagonist leaves the binding pocket once the biotin has been attached to the lysines. It might be helpful if the authors made this a little more apparent in Figure 1 or in the text describing the NASA chemistry reaction.

We thank the reviewer for this insightful suggestion. To address this, we have modified Figure 1A and updated the legend.

**Reviewer #3 (Public review):**
Summary:This manuscript describes the development of a covalent labeling probe (X7-uP) that selectively targets and tags native P2X7 receptors at the plasma membrane of BV2 microglial cells. Using super-resolution imaging (dSTORM), the authors demonstrate that P2X7 receptors form nanoscale clusters upon microglial activation by lipopolysaccharide (LPS) and ATP, correlating with synergistic IL-1β release. These findings advance understanding of P2X7 reorganization during inflammation and provide a generalizable labeling strategy for monitoring endogenous P2X7 in immune cells.Strengths:(1) The authors designed X7-uP by coupling a high-affinity, P2X7-specific antagonist (AZ10606120) with N-cyanomethyl NASA chemistry to achieve site-directed biotinylation. This approach offers high specificity, minimal off-target reactivity, and a straightforward pull-down/imaging readout.(2) The results connect P2X7's nanoscale clustering directly with IL-1β secretion in microglia, reinforcing the role of P2X7 in inflammation. By localizing endogenous P2X7 at single-molecule resolution, the authors reveal how LPS priming and ATP stimulation synergistically reorganize the receptor.(3) The authors systematically validate their method in recombinant systems (HEK293 cells) and in BV2 cells, showing selective inhibition, mutational confirmation of the binding site, and Western blot pulldown experiments.

We thank the reviewer for their positive comment.

Weaknesses:(1) While the data strongly indicate that P2X7 clustering contributes to IL-1β release, the manuscript would benefit from additional experiments (if feasible) or discussion on how receptor clustering interfaces with downstream inflammasome assembly. Clarification of whether the P2X7 clusters physically colocalize with known inflammasome proteins would solidify the mechanism.

We thank the reviewer for this valuable suggestion. Determining the physical colocalization of P2X7 clusters with known inflammasome components would provide important insight into the molecular partners involved in inflammasome activation. However, we believe that such an investigation would constitute a substantial study on its own and therefore lies beyond the scope of the present work.

Nevertheless, in response to the reviewer’s suggestion, we have added a short paragraph at the end of the Discussion section addressing potential mechanisms by which P2X7 clustering may contribute to downstream inflammasome activation. We also revised the text to tone down the hypothesis of physical colocalization.

(2) The authors might expand on the scope of X7-uP in other native cells that endogenously express P2X7 (e.g., macrophages, dendritic cells). Although they mention the possibility, demonstrating the probe's applicability in at least one other primary immune cell type would strengthen its general utility.

We thank the reviewer for this valuable suggestion. Again, we believe that such an investigation would constitute a substantial study on its own and therefore lies beyond the scope of the present work.

(3) The authors do include appropriate negative controls, yet providing additional details (e.g., average single-molecule on-time or blinking characteristics) in supplementary materials could help readers assess cluster calculations.

As suggested, we have included additional data showing single-molecule blinking events in untreated and LPS+ATP-treated BV2 cells, along with the corresponding movies. The data are now presented in Figure 5—supplement figure 3A and B and Figure 5—Videos 1 and 2.

**Recommendations for the authors:**

**Reviewer #2 (Recommendations for the authors):**
(1) On line 96, the authors refer to the "ballast" domain of P2X7 receptor but do not cite the original article from which this nomenclature originated (McCarthy et al., 2019, Cell). This article should be cited to give appropriate credit.

Done.

(2) On line 602, the authors state that they use models from PDB 1MK5 and 6U9W to generate the cartoons in Figure 6. The manuscripts from which these PDB files were generated need to be appropriately cited.

Done.

(3) On line 319, the authors say "300 mM BzATP" but I think they mean 300 uM.

Done. Thank you for catching the typo.

**Reviewer #3 (Recommendations for the authors):**
Overall, excellent data quality. The paper would benefit from a discussion of the physiological implications of clustering. It would also be helpful to elaborate about the potential mechanisms for clustering: diffusion and/or insertion. Finally, the authors should comment on work by Mackinnon's (PMID: 39739811) and Santana lab (PMID: 31371391) on two distinct models for clustering of proteins.

As suggested by the reviewer, we have revised the discussion to incorporate their comments. First, we have added the following text:

“Upon BV2 activation, we observed significant nanoscale reorganization of P2X7. Both LPS and ATP (or BzATP) trigger P2X7 upregulation and clustering, increasing the overall number of surface receptors and the number of receptors per cluster, from one to three (Figure 6). By labeling BV2 cells with X7-uP shortly after IL-1b release, we were able to correlate the nanoscale distribution of P2X7 with the functional state of BV2 cells, consistent with the two-signal, synergistic model for IL-1b secretion observed in microglia and other cell types (Ferrari et al, 1996; Perregaux et al, 2000; Ferrari et al, 2006; Di Virgilio et al, 2017; He et al, 2017; Swanson et al, 2019). In this model, LPS priming leads to intracellular accumulation of pro-IL-1b, while ATP stimulation activates P2X7, triggering NLRP3 inflammasome activation and the subsequent release of mature IL-1b.

What is the mechanism underlying P2X7 upregulation that leads to an overall increase in surface receptors—does it result from the lateral diffusion of previously masked receptors already present at the plasma membrane, or from the insertion of newly synthesized receptors from intracellular pools in response to LPS and ATP? Although our current data do not distinguish between these possibilities, a recent study suggests that the a1 subunit of the Na^+^/K+-ATPase (NKAa1) forms a complex with P2X7 in microglia, including BV2 cells, and that LPS+ATP induces NKAa1 internalization (Huang et al, 2024). This internalization appears to release P2X7 from NKAa1, allowing P2X7 to exist in its free form. We speculate that the internalization of NKAa1 induced by both LPS and ATP exposes previously masked P2X7 sites, including the allosteric AZ10606120 sites, thus making them accessible for X7-uP labeling.”

Second, we have added a short paragraph at the end of the Discussion section addressing potential mechanisms by which P2X7 clustering may contribute to downstream inflammasome activation:

“What mechanisms underlie P2X7 clustering in response to inflammatory signals? Several models have been proposed to explain membrane protein clustering, including recruitment to structural scaffolds (Feng & Zhang, 2009), partitioning into membrane domains enriched in specific chemical components such as lipid rafts (Simons & Ikonen, 1997), and self-assembly mechanisms (Sieber et al, 2007). These self-assembly mechanisms include an irreversible stochastic model (Sato et al, 2019) and a more recent reversible self-oligomerization model which gives rise to higher-order transient structures (HOTS) (Zhang et al, 2025). Supported by cryogenic optical localization microscopy with very high resolution (~5 nm), the HOTS model has been observed in various membrane proteins, including ion channels and receptors (Zhang et al, 2025). Furthermore, HOTS are suggested to be dynamically modulated and to play a functional role in cell signaling, potentially influencing both physiological and pathological processes (Zhang & MacKinnon, 2025). While this hypothesis is compelling, our current dSTORM data lack sufficient spatial resolution to confirm whether P2X7 trimers form HOTS via self-oligomerization. Further biophysical and ultra-high-resolution imaging studies are required to test this model in the context of P2X7 clustering.”